The cranial anatomy of the neornithischian dinosaur Thescelosaurus neglectus

Boyd Clint A. clintboyd@stratfit.org
Department of Geology and Geological Engineering Sciences, South Dakota School of Mines and Technology , Rapid City, SD , USA
Farke Andrew
Electronic publication date: 2014 Nov 13
Publication date: 2014
Volume: 2
Electronic Location ID: e669
Received 2014 Aug 7; Accepted 2014 Oct 28
Copyright: © 2014 Boyd
Copyright year: 2014
Copyright holder: Boyd
License: This is an open access article distributed under the terms of the Creative Commons Attribution License, which permits unrestricted use, distribution, reproduction and adaptation in any medium and for any purpose provided that it is properly attributed. For attribution, the original author(s), title, publication source (PeerJ) and either DOI or URL of the article must be cited.
License URL: https://creativecommons.org/licenses/by/4.0/

Keywords: Neornithischia, Ornithischia, Dinosauria, Maastrichtian, Thescelosaurus, Hell Creek Formation

Funding: The American Museum of Natural History Collection Study Grant Ernest L. and Judith W. Lundelius Scholarship Francis L. Whitney Endowed Presidential Scholarship Geological Society of America Graduate Student Research Grant National Science Foundation The North Carolina Fossil Club Jackson School of Geosciences at the University of Texas at Austin Department of Marine, Earth, and Atmospheric Sciences at North Carolina State University This research was funded via the American Museum of Natural History Collection Study Grant, an Ernest L. and Judith W. Lundelius Scholarship in Vertebrate Paleontology, a Francis L. Whitney Endowed Presidential Scholarship from the University of Texas, a Geological Society of America Graduate Student Research Grant, the National Science Foundation’s East Asia and Pacific Summer Institutes for U.S. Graduates Students program, the North Carolina Fossil Club, and financial support from the Jackson School of Geosciences at the University of Texas at Austin and the Department of Marine, Earth, and Atmospheric Sciences at North Carolina State University. The funders had no role in study design, data collection and analysis, decision to publish, or preparation of the manuscript.

==============================
Though the dinosaur Thescelosaurus neglectus was first described in 1913 and is known from the relatively fossiliferous Lance and Hell Creek formations in the Western Interior Basin of North America, the cranial anatomy of this species remains poorly understood. The only cranial material confidently referred to this species are three fragmentary bones preserved with the paratype, hindering attempts to understand the systematic relationships of this taxon within Neornithischia. Here the cranial anatomy of T. neglectus is fully described for the first time based on two specimens that include well-preserved cranial material (NCSM 15728 and TLAM.BA.2014.027.0001). Visual inspection of exposed cranial elements of these specimens is supplemented by detailed CT data from NCSM 15728 that enabled the examination of otherwise unexposed surfaces, facilitating a complete description of the cranial anatomy of this species. The skull of T. neglectus displays a unique combination of plesiomorphic and apomorphic traits. The premaxillary and ‘cheek’ tooth morphologies are relatively derived, though less so than the condition seen in basal iguanodontians, suggesting that the high tooth count present in the premaxillae, maxillae, and dentaries may be related to the extreme elongation of the skull of this species rather than a retention of the plesiomorphic condition. The morphology of the braincase most closely resembles the iguanodontians Dryosaurus and Dysalotosaurus, especially with regard to the morphology of the prootic. One autapomorphic feature is recognized for the first time, along with several additional cranial features that differentiate this species from the closely related and contemporaneous Thescelosaurus assiniboiensis. Published phylogenetic hypotheses of neornithischian dinosaur relationships often differ in the placement of the North American taxon Parksosaurus, with some recovering a close relationship with Thescelosaurus and others with the South American taxon Gasparinisaura, but never both at the same time. The new morphological observations presented herein, combined with re-examination of the holotype of Parksosaurus, suggest that Parksosaurus shares a closer relationship with Thescelosaurus than with Gasparinisaura, and that many of the features previously cited to support a relationship with the latter taxon are either also present in Thescelosaurus, are artifacts of preservation, or are the result of incomplete preparation and inaccurate interpretation of specimens. Additionally, the overall morphology of the skull and lower jaws of both Thescelosaurus and Parksosaurus also closely resemble the Asian taxa Changchunsaurus and Haya, though the interrelationships of these taxa have yet to be tested in a phylogenetic analysis that includes these new morphological data for T. neglectus.

Introduction

Thescelosaurus neglectus is a relatively large-bodied ‘hypsilophodontid’ taxon (adult size >4 m: Fisher et al., 2000) known only from the late Maastrichtian of North America (Norman et al., 2004; Boyd et al., 2009). The holotype (USNM 7757) and paratype (USNM 7758) were each collected from sediments of the Lance Formation exposed in Niobrara County, Wyoming (Gilmore, 1913). While the holotype consists of a relatively complete postcranial skeleton, the paratype is highly fragmentary and was selected because it preserved portions of the forelimb not present in the holotype (Boyd et al., 2009: Fig. 2). A full description of T. neglectus based on these and other specimens was published by Gilmore in 1915 and although the anatomy of nearly the entire postcranial skeleton was described, no portion of the skull was recognized at that time (Gilmore, 1915).

The cranial anatomy of T. neglectus was completely unknown until 1974 when Thescelosaurus edmontonensis (holotype = CMN 8537) from the Scollard Formation of Alberta, Canada was subjectively synonymized with T. neglectus (Galton, 1974b). The holotype and only specimen of T. edmontonensis includes the frontals, parietal, left postorbital, right prootic, supraoccipital, fused left opisthotic/exoccipital, and an articulated left lower jaw missing the coronoid and predentary (Boyd et al., 2009: Fig. 2). At the same time, an isolated, toothless dentary (AMNH 5052) from the Hell Creek Formation of Montana was referred to T. neglectus based on its similarity to that of CMN 8537 (Galton, 1974b). Shortly thereafter, a specimen from the Hell Creek Formation of Montana (LACM 33543) was referred to T. neglectus (Morris, 1976). That specimen also preserves portions of the skull, including a partial braincase (Boyd et al., 2009: Fig. 2), but the presence of two right jugals indicates this material represents at least two individuals (Morris, 1976). A fourth specimen (RSM P 1255.1) was later referred to T. neglectus from the Frenchman Formation of Saskatchewan, Canada that preserves a partial skull, including a relatively complete braincase (Galton, 1989). These four specimens formed the basis for a detailed description and reconstruction of the skull of T. neglectus (Galton, 1997: Figs. 3G and 3H). However, these referrals were not based on shared apomorphies, but on general similarity. Given the lack of comparative cranial material recognized from the holotype and paratype of T. neglectus and the presence of only a single postcranial character distinguishing T. neglectus from Thescelosaurus garbanii, the latter of which is only known from a partial hindlimb and some associated vertebrae (Morris, 1976), the referral of all of these specimens to T. neglectus at that time was tenuous at best.

Discovery of previously unrecognized cranial material preserved with the paratype of T. neglectus (partial left frontal, left postorbital, and left squamosal) spurred a taxonomic revision of all significant ‘hypsilophodontid’ specimens (i.e., type specimens or relatively complete skeletons) from the Maastrichtian of North America (Boyd et al., 2009). That study recognized the presence of four diagnosably distinct ‘hypsilophodontid’ species: Parksosaurus warreni; T. garbanii; T. neglectus; and, an unnamed species of Thescelosaurus represented by RSM P 1225.1 (now the holotype of Thescelosaurus assiniboiensis). That study also concluded that all other specimens previously referred to Thescelosaurus could only be referred to Thescelosaurus incertae sedis owing to the inability to compare those specimens to the type material of all three recognized species (Boyd et al., 2009). As a result, the cranial description and reconstruction of T. neglectus provided by Galton (1997: Figs. 3G and 3H) is based on specimens that are either referable to a separate species (i.e., RSM P 1225.1) or on specimens that currently cannot be identified to the species level, reducing our knowledge of the cranial anatomy of T. neglectus to only that material preserved with the paratype.

NCSM 15728 was collected in 1999 from Hell Creek Formation sediments in Harding County, South Dakota. This specimen includes much of the axial skeleton, part of the appendicular skeleton (largely from the right side), and a three-dimensionally preserved skull missing only part of the left quadratojugal (Fig. 1). Despite the excellent condition of this specimen and the poor understanding of the cranial anatomy of Thescelosaurus, prior research on NCSM 15728 focused on the possible preservation of soft tissue structures in the specimen (Fisher et al., 2000; Rowe, McBride & Sereno, 2001; Russell et al., 2001; Cleland, Stoskopf & Schweitzer, 2011) and the histology, morphology, and osteogenesis of de novo ossifications associated with the anterior dorsal ribs (Boyd, Cleland & Novas, 2011). NCSM 15728 was originally referred to Thescelosaurus neglectus by Fisher et al. (2000) based on general similarity to the types, and Boyd et al. (2009) noted that the cranial morphology of NCSM 15728 was consistent with the paratype of T. neglectus and distinct from the holotype of T. assiniboiensis. However, Boyd et al. (2009) referred NCSM 15728 to Thescelosaurus incertae sedis because it could not be sufficiently compared to the type material of T. garbanii (LACM 33542).

Figure 1 Skull of NCSM 15728 in right lateral view.

(A) diagram highlighting the contacts between the bones on the right side of skull; (B) illustration of right side of skull; (C) photograph of right side of skull. In (A) and (B), grey regions indicate the presence of matrix on the specimen. Abbreviations: an, angular; asor, accessory supraorbital; bo, basioccipital; de, dentary; eo, fused opisthotic/exoccipital; fr, frontal; ju, jugal; la, lacrimal; mx, maxilla; na, nasal; pd, predentary; pf, prefrontal; pm, premaxilla; po, postorbital; pop, paroccipital process; qj, quadratojugal; qu, quadrate; sor, supraorbital; sq, squamosal; su, surangular. Scale bars equal 10 cm.

Subsequent examination of a previously unreported specimen of Thescelosaurus (TLAM.BA.2014.027.0001) collected from Hell Creek Formation sediments in Dewey County, South Dakota facilitated indirect comparison of NCSM 15728 to the holotype of T. garbanii. These comparisons support the confident referral of both NCSM 15728 and TLAM.BA.2014.027.0001 to T. neglectus. These referrals and the excellent preservation of the skull of NCSM 15728 allows the cranial anatomy of T. neglectus to be fully described for the first time since the species was named a century ago. This description is based on personal observations of the exposed portions of the skulls of NCSM 15728 and TLAM.BA.2014.027.0001 and the use of computed tomography (CT) technology to image and reconstruct the unexposed portions of the skull of NCSM 15728, providing insights into portions of the cranial anatomy of basal neornithischians that were previously unknown or poorly understood. A new diagnosis for T. neglectus is presented that clearly distinguishes this species from all known basal ornithischian and basal ornithopod taxa. The data presented herein are crucial for gaining a clearer understanding of the evolution of the skull in neornithischian dinosaurs and for assessing the systematic relationships not only of the taxon Thescelosaurus, but for all neornithischian dinosaurs.

Materials & Methods

The anatomy of NCSM 15728 was studied using a combination of methodologies that provided maximum insight into the cranial morphology of Thescelosaurus neglectus. Initial preparation of the skull was conducted by Michael Hammer, who discovered and excavated the specimen. This initial phase of preparation focused on exposing the right lateral side of the skull, portions of the dorsal and posterior surfaces, and some of the left lateral surface to ready the specimen for exhibition. Additional preparation work was conducted on the skull of NCSM 15728 under the direction and with the assistance of Dr. Paul Brinkman (NCSM). This second phase of preparation focused on removing matrix from the dorsal surface of the parietal, inside the supratemporal fenestrae, the entire posterior surface of the skull, within the left orbit and antorbital fenestra, within the nares, ventrally between the lower jaws, and between the oral margins of the premaxillae and predentary. The left quadratojugal, the posterior three-quarters of the jugal, and the left quadrate (not including the proximal head) were removed, exposing the lateral surfaces of the posterior palatal elements and the braincase (Fig. 2). The anatomical data gleaned from personal observations of the exposed surfaces of NCSM 15728 were supplemented by computed tomography (CT) scans of the skull, not including the elements removed from the left side of the skull. The CT scans were conducted at the College of Veterinary Medicine at North Carolina State University using a Siemens Somatom Sensation 16. The slice thickness is 0.75 mm, the interslice spacing is 0.0 mm, and the voxel size is 0.414 mm by 0.414 mm by 1.000 mm (Cleland, Stoskopf & Schweitzer, 2011; T Cleland, pers. comm., 2014). The final dataset consists of 300 DICOM files. Digital models of some of the bones of the cranium that could not be described adequately via visual examination of the specimen (e.g., bones of the palate) were constructed using the program VGSudio Max in the digital morphology lab at The University of Texas at Austin. These CT data provided insight into areas of the skull that cannot be observed directly owing to the presence of matrix on the specimen that was retained for structural support and the manner in which the specimen was mounted for display. The combination of these methods ensures that the elucidation of the anatomy of this specimen is only limited by the preservation of the specimen. These CT data are reposited in the Digital Morphology library at the University of Texas at Austin and are available upon request.

Figure 2 Skull of NCSM 15728 in left lateral view.

(A) diagram highlighting the contacts between the bones on the left side of skull; (B) illustration of left side of skull; (C) photograph of left side of skull. In (A) and (B), grey regions indicate the presence of matrix on the specimen. Abbreviations: an, angular; bo, basioccipital; de, dentary; eo, fused opisthotic/exoccipital; fr, frontal; ju, jugal; la, lacrimal; mx, maxilla; na, nasal; par, parietal; pd, predentary; pf, prefrontal; pl, palatine; pm, premaxilla; po, postorbital; pop, paroccipital process; pro, prootic; ps, parasphenoid; pt, pterygoid; qu, quadrate; so, supraoccipital; sp, sclerotic plate; sq, squamosal; st, stapes; su, surangular. Scale bars equal 10 cm.

The anatomy of TLAM.BA.2014.027.0001 was studied via personal examination of the exposed areas of the skull and the associated, but disarticulated, right quadrate and fused right exoccipital/opisthotic. Initial preparation of this specimen was largely conducted by Bill Alley, who discovered and excavated the specimen from private lands and later donated the specimen to the Timber Lake and Area Museum. Additional preparation of the skull of TLAM.BA.2014.027.0001 was conducted by the author at the Paleontological Research Laboratory at the South Dakota School of Mines and Technology to remove sediment and previously applied consolidants from all of the exposed surfaces of the skull.

Systematic Paleontology

The systematic position of Thescelosaurus neglectus, and all former ‘hypsilophodontids’ in general, within Ornithischia remains hotly debated, which creates difficulties when selecting appropriate clade names to apply when discussing this taxon. Thescelosaurus neglectus was originally thought to be closely related to basal ankylopollexians (e.g., Camptosaurus dispar) within Ornithopoda, based on a preliminary examination of the hypodigm material (Gilmore, 1913), but was soon after referred to the Hypsilophodontidae (Gilmore, 1915). That referral was upheld by most subsequent authors for more than sixty years (e.g., Parks, 1926; Swinton, 1936; Janensch, 1955; Romer, 1956; Romer, 1966; Thulborn, 1970; Thulborn, 1972), with a few notable exceptions. Sternberg (1940) placed T. neglectus in its own clade within Hypsilophodontidae, which he named Thescelosaurinae (= Thescelosauridae of Sternberg (1937)), a referral that was followed by some authors (e.g., Kuhn, 1966; Morris, 1976). Galton (1971a), Galton (1971b), Galton (1972), Galton (1973) and Galton (1974b) argued against the placement of T. neglectus within Thescelosaurinae and even Hypsilophodontidae, instead referring the taxon to Iguanodontidae. Galton (1995), Galton (1997) and Galton (1999) later reassessed that referral and instead assigned T. neglectus to the Hypsilophodontidae. Despite these taxonomic disagreements, the placement of T. neglectus within Ornithopoda (sensu Butler, Upchurch & Norman, 2008) was uncontested by all these authors.

The relatively recent recognition of Hypsilophodontidae as a paraphyletic set of taxa (e.g., Scheetz, 1999; Butler, Upchurch & Norman, 2008; Boyd et al., 2009; Brown, Boyd & Russell, 2011) raised the question of whether all former ‘hypsilophodontids’ belong within Ornithopoda (sensu Butler, Upchurch & Norman, 2008), or if some of those taxa are non-cerapodan, basal neornithischians (sensu Butler, Upchurch & Norman, 2008). Unfortunately, most recent phylogenetic analyses have provided little resolution regarding the postion of T. neglectus within Neornithischia relative to the clade Ornithopoda for a variety of reasons. Several analyses that included T. neglectus did not include any marginocephalian taxa, making it impossible to determine if T. neglectus is placed within a monophyletic Ornithopoda (Weishampel & Heinrich, 1992; Scheetz, 1999; Varricchio, Martin & Katsura, 2007; Boyd et al., 2009). Furthermore, the strict consensus trees produced by Butler (2005), Spencer (2007), and Butler, Upchurch & Norman (2008) placed T. neglectus in a large polytomy within Neornithischia, precluding its definitive referral to Ornithopoda. Another published study (Buchholz, 2002) presented only one of the most parsimonious trees recovered during the analysis, making it impossible to determine if T. neglectus was recovered within Ornithopoda in all ten of the recovered most parsimonious trees. Additionally, other analyses have a priori assumed the inclusion of the T. neglectus within Ornithopoda and used the sister taxon of Ornithopoda, Marginocephalia, as an outgroup, ensuring that T. neglectus was recovered within Ornithopoda (e.g., Weishampel et al., 2003). Thus, in no previous phylogenetic analysis of ornithischian relationships was T. neglectus unambiguously recovered within Ornithopoda when its position within Neornithischia was thoroughly assessed.

As a result of these disparate hypotheses regarding the systematic relationships of T. neglectus, various clade names have been used and are still used to refer to both this taxon and former ‘hypsilophodontids’ in general. The terms most commonly used to refer to these taxa are ‘hypsilophodontid’ and basal ornithopod. The former term should be avoided because it refers to a paraphyletic grade of ornithischian dinosaurs and does not provide precise information regarding the relationships of the taxon or taxa in question. The latter term is too precise, giving the inaccurate impression that the position of T. neglectus specifically, and ‘hypsilophodontids’ in general, within Ornithopoda is certain, when to date the evidence is ambiguous. Alternatively, Boyd et al. (2009) referred to all former ‘hypsilophodontids’ as basal neornithischians to reflect that the least inclusive group these taxa have been definitively referred to is Neornithischia and that their various postions within that clade (i.e., within or outside of Ornithopoda) remain uncertain. However, Butler, Upchurch & Norman (2008) used the term basal neornithischian to more precisely refer to taxa recovered within Neornithischia but definitively positioned outside of Cerapoda. Thus, the application of the term basal neornithischian by Butler, Upchurch & Norman (2008) is preferred over the usage by Boyd et al. (2009). In the present study, T. neglectus and all other taxa definitively placed within Neornithischia but outside of both Marginocephalia and Iguanodontia, including all former ‘hypsilophodontids,’ are conservatively referred to simply as neornithischians, which requires no inference as to whether or not some or all of these taxa are also ornithopods.

DINOSAURIA Owen, 1842	
ORNITHISCHIA Seeley, 1887	
NEORNITHISCHIA Cooper, 1985 (sensu Butler, Upchurch & Norman, 2008)	
THESCELOSAURUS Gilmore, 1913	
Bugenasaura Galton, 1995:308	

Name bearing species

Thescelosaurus neglectus Gilmore, 1913

Other included species

Thescelosaurus garbanii Morris, 1976

Thescelosaurus assiniboiensis Brown, Boyd & Russell, 2011

Distribution

Frenchman Formation, Saskatchewan; Hell Creek Formation, Montana, North Dakota, and South Dakota; Lance Formation, Wyoming; Scollard Formation, Alberta (all Maastrichtian age [72.1–66.0 Ma]; Weishampel et al., 2004; Cohen et al., 2013).

Diagnosis

The following apomorphies distinguish Thescelosaurus from all other basal ornithischian dinosaurs (Boyd et al., 2009; Brown, Boyd & Russell, 2011): (1) Frontals wider at midorbital level than across posterior end; (2) dorsolaterally directed process on surangular; (3) prominent, horizontal ridge on maxilla with at least the posterior portion covered by a series of coarse, rounded, obliquely inclined ridges; (4) depressed posterior half of ventral edge of jugal covered laterally with obliquely inclined ridges; (5) foramen in dorsal surface of prefrontal that opens into the orbit positioned dorsomedial to the articulation surface for palpebral; and (6) shafts of anterior dorsal ribs transversely compressed and laterally concave, with the posterior margin of the distal half characterized by a distinct rugose texture and flattened surface, possibly for articulation with the intercostal plates. Two additional characters are currently uniquely known in Thescelosaurus, but are unable to be evaluated in its recovered sister taxon Parksosaurus (Boyd et al., 2009): (1) dorsal edge of opisthotic indented by deep, ‘Y-shaped’ excavation in dorsal view; and, (2) palpebral dorsoventrally flattened and rugose along the medial and distal edges.

Two additional characters are optimized as local apomorphies of Thescelosaurus, but occur convergently within major neornithischian subclades: (1) angle between ventral margin of braincase (occipital condyle, basal tubera, and basipterygoid processes) and a line drawn through center of the trigeminal foramen and posterodorsal hypoglossal foramen less than fifteen degrees and (2) femur longer than tibia. The former also is found in some iguanodontians (e.g., Tenontosaurus: Norman, 2004) and the latter occurs in some iguanodontians and marginocephalians (Maryańska, Chapman & Weishampel, 2004; Norman, 2004).

THESCELOSAURUS NEGLECTUS (Gilmore, 1913)

Holotype

USNM 7757: nearly complete postcranial skeleton.

Paratype

USNM 7758: fragmentary skeleton including parts of skull.

Type series localities

USNM 7757: Collected by JB Hatcher and WH Utterback in 1891 from Doegie Creek, Niobrara County, Wyoming. USNM 7758: Collected by OA Peterson in 1889 from Lance Creek, Niobrara County, Wyoming.

Distribution

Lance Formation of Wyoming and Hell Creek Formation of South Dakota (both Maastrichtian age [72.1–66.0 Ma] (Weishampel et al., 2004; Cohen et al., 2013)).

Referred specimens

NCSM 15728 (Figs. 1–19; Table 1): Complete skull and lower jaws (lacking only part of the left quadratojugal), ceratobranchials, articulated vertebral column complete from the atlas to the thirteenth caudal vertebra, cervical, dorsal, and sternal ribs, seven right intercostal plates, nine chevrons, right fused scapulocoracoid, left and right sternal plates, right humerus, right ulna, right radius, right manus consisting of five carpals, all five metacarpals, and seven phalanges, right ilium, left and right pubes, left and right ischia, right femur, proximal portion of the right tibia, proximal half of the right fibula.

Table 1 Selected measurements of specimens referred to Thescelosaurus neglectus.

All measurements in mm. Premaxilla length measured perpendicular to the oral margin of the premaxilla from the anterior-most tip to the posterior extent of the posterolateral processes. Maxillary tooth row length measured in straight line from the posterior-most alveolus to the anterior-most alveolus. Dentary length is maximum total length of the dentary in lateral view. Predentary length measured as maximum length of the oral margin of the predentary. Frontal width measured at widest point across a single frontal in a line perpendicular to the midline suture. Quadrate height is the maximum height of the quadrate from the distal condyles to the dorsal head.

Specimen number	Premaxilla
length	Maxillary tooth
row length	Dentary
length	Predentary
length	Frontal
width	Quadrate
height	
NCSM 15728	101.4	90.4	146.7	57.4	33.2	94.9	
TLAM.BA.2014.027.0001	–	87.2	–	–	29.1	87.2	

TLAM.BA.2014.027.0001 (Figs. 8C, 8D; Table 1): Relatively complete, slightly transversely crushed skull missing the supraorbitals, accessary supraorbitals, right postorbital, right quadratojugal, most of the right jugal (anterior-most process preserved), and the entire lower jaws. Preserved postcranial elements consist of the left antlantal neural arch, the atlantal intercentrum, eight dorsal vertebrae, the sacrodorsal, three sacral vertebrae (all unfused), forty-three caudals, six partial dorsal ribs, nine chevrons, left scapulocoraoid, partial right and left pubes, right ischium, partial proximal right tibia, incomplete distal ends of right and left tibiae, partial right astragulus, right calcaneum, distal ends of right metatarsals III and IV, eight pedal phalanges, and additional unidentified material.

Figure 3 Skull of NCSM 15728 in dorsal view.

(A) diagram highlighting the contacts between the bones on the dorsal surface of skull; (B) illustration of dorsal surface of skull; (C) photograph of dorsal surface of skull. In (A) and (B), grey regions indicate the presence of matrix on the specimen. Abbreviations: asor, accessory supraorbital; eo, fused opisthotic/exoccipital; fr, frontal; ju, jugal; mx, maxilla; na, nasal; par, parietal; pf, prefrontal; pm, premaxilla; po, postorbital; pop, paroccipital process; qj, quadratojugal; qu, quadrate; so, supraoccipital; sor, supraorbital; sp, sclerotic plates; sq, squamosal. Scale bars equal 10 cm.

Figure 4 Skull of NCSM 15728 in posterior view.

(A) diagram highlighting the contacts between the bones on the posterior side of skull; (A) illustration of posterior side of skull; (B) photograph of posterior side of skull. In (A) and (B), grey regions indicate the presence of matrix on the specimen. Abbreviations: an, angular; ar, articular; bo, basioccipital; eo: fused opisthotic/exoccipital; fr, frontal; par, parietal; po, postorbital; pop, paroccipital process; pro, prootic; qu, quadrate; so, supraoccipital; sq, squamosal; su, surangular. Scale bars equal 10 cm.

Figure 5 Premaxillae of NCSM 15728.

(A) right premaxilla in lateral view; (B) premaxillae in dorsal view; (C) posterior portion of the left premaxillary palate in ventrolateral view; (D) external nares in left lateral view; (E) anterior portion of the premaxillary palate. In (A), (B), (D), and (E) the directional arrows indicate the orientation of the specimen. In (C), anterior is to the left. Dashed white line in (C) indicates the shape and position of the contact between the vomer and the premaxilla. Abbreviations: ads, anterodorsal shelf of premaxilla; amf, anterior maxillary fossa; almp, anterolateral maxillary process; apmf, anterior premaxillary foramen; avp, anteroventral tip of premaxilla; de, dentary, dt, dentary tooth/teeth; lat, lateral; med, medial, mt, maxillary tooth/teeth; mx, maxilla; na, nasal; pd, predentary; pdp, posterodorsal process of the premaxilla; plp, posterolateral process of premaxilla; pls, posterolateral sulcus in premaxilla; pm, premaxilla; pmf, premaxillary foramen; pmt, premaxillary tooth/teeth; pnp, premaxillary narial process; post, posterior; rpf, rostral palatal foramen; vent, ventral; vo, vomer. Scale bars equal 1 cm.

Figure 6 Lacrimal and prefrontal of NCSM 15728.

(A) right lacrimal in lateral view; (B) left lacrimal in posterolateral view; (C) left prefrontal in lateral view (note: ventral process not shown because it was obscured by the lacrimal); (D) left prefrontal in dorsal view. The directional arrows indicate the orientation of the specimen in each view. Abbreviations: ant, anterior; aso, articulation for supraorbital; dor, dorsal; drmm, dorsal rim of the medial process of the maxilla; eaof, external antorbital fenestra; iaof, internal antorbital fenestra; ju, jugal; la, lacrimal; laa, articulation surface for lacrimal; lf, lacrimal foramen; med, medial; mx, maxilla; na, nasal; or, orbit; pf, prefrontal; pfa, prefrontal articulation surface; pff, prefrontal foramen; post, posterior; rso, rugose contact for supraorbital; sor, supraorbital; vent, ventral. Scale bars equal 1 cm.

Figure 7 Jugal and postorbital of NCSM 15728.

(A) partial left jugal in lateral view; (B) partial left jugal in medial view; (C) right postorbital in lateral view. The directional arrows indicate the orientation of the specimen in each view. Abbreviations: aip, anterior inflation of postorbital; apo, articulation surface for postorbital; app, anterior process of postorbital; asq, articulation surface for squamosal; dor, dorsal; dpj, dorsal projection of posterior process of jugal; fr, frontal; lp, lateral process of posterior process of postorbital; mgj, medial groove on jugal; mp, medial projection of the posterior process of the postorbital; par, parietal; po, postorbital; post, posterior; pro, prootic; ps, parasphenoid; sed, sediment; soaa, articulation surface for accessory supraorbital; so, supraoccipital; sp, sclerotic plate; sq, squamosal; vd, ventral depression on jugal; vent, ventral; vpj, ventral process of the posterior projection of the jugal; vppo, ventral process of the postorbital. Scale bars equal 1 cm.

Figure 8 Squamosal and quadrate of Thescelosaurus neglectus

(A) left squamosal of NCSM 15728 in lateral view; (B) right squamosal of NCSM 15728 in dorsal view; (C) right quadrate of TLAM.BA.2014.027.0001 in medial view; (D) right quadrate of TLAM.BA.2014.027.0001 in lateral view; (F) close up of the pterygoid wing on the left quadrate of NCSM 15728; (F) contact between the left squamosal and postorbital of NCSM 15728 in dorsal view; (G) foramen between the right quadrate and quadratojugal of NCSM 15728 in posterolateral view. The directional arrows indicate the orientation of the specimen in each view. Abbreviations: ant, anterior; aqj, articulation for quadratojugal; asq, articulation surface for squamosal; c, concretion; d, damage; dor, dorsal; dsq, dorsal projection of the anterior process of squ amosal; ju, jugal; jw, jugal wing; lat, lateral; lp, lateral process of posterior process of postorbital; med, medial; mp, medial projection of the posterior process of the postorbital; po, postorbital; pop, paroccipital process; poq, postquadratic process of squamosal; post, posterior; prq, prequadratic process; pw, pterygoid wing; pwf, pterygoid wing fossa; pwg, pterygoid wing ventral groove; qf, quadrate foramen; qj, quadratojugal; qu, quadrate; rsq, ventral ridge on squamosal; sed, sediment; so, supraoccipital; sq, squamosal; vent, ventral; vsq, ventral projection of the anterior process of the squamosal. Scale bars equal 1 cm.

Figure 9 Midline and left palatal elements of NCSM 15728 derived from CT scans.

(A) left palatal elements in lateral view; (B) left palatal elements in medial view; (C) left palatal elements in dorsal view; (D) left palatal elements in ventral view; (E) left pterygoid in lateral view; (F) left pterygoid in medial view; (G) left pterygoid in dorsal view; (H) vomer in left lateral view; (I) vomer in dorsal view. Key to colors used in (A) through (D): Red, Palatine; Green, Pterygoid; Yellow, Ectopterygoid; Orange, Vomer. The directional arrows indicate the orientation of the specimen in each view. Abbreviations: ant, anterior; apa, articulation for palatine; bpa, basipterygoid articulation; d, damage; dor, dorsal; ea, ectopterygoid articulation; lat, lateral; lpr, lateral pterygoid ridge; med, medial; mpp, mandibular process of pterygoid; pg, pterygoid groove; post, posterior; ppp, palatine process of pterygoid; ppv, posterior process of vomer; qap, quadrate alar process; qpp, quadrate process of pterygoid; vent, ventral. In (A) through (D) scale bars equal 5 cm. In (E) through (I) scale bars equal 1 cm.

Figure 10 Additional illustrations of left palatal elements of NCSM 15728 derived from CT scans.

(A) left palatine in lateral view; (B) left palatine in anterior view; (C) left palatine in medial view; (D) left ectopterygoid in dorsal view; (E) left ectopterygoid in posterior view; (F) left ectopterygoid in anterior view. The directional arrows indicate the orientation of the specimen in each view. Abbreviations: adp, anterodorsal process of palatine; aj, articulation for jugal; am, articulation for maxilla; ant, anterior; apa, articulation for palatine; apt, articulation for pterygoid; dor, dorsal; lat, lateral; med, medial; post, posterior; ppf, postpalatine fenestra; vent, ventral. Scale bars equal 1 cm.

Figure 11 Midline and left side elements of the braincase of NCSM 15728 derived from CT scans.

(A) braincase in lateral view; (B) braincase in medial view; (C) braincase in dorsal view; (D) braincase in ventral view; (E) braincase in posterior view; (F) braincase in anterior view. Key to colors: Red, Prootic; Yellow, Laterosphenoid; Purple, Supraoccipital; Green, Basioccipital; Blue, Fused basisphenoid/parasphenoid; Orange, Fused opisthotic/exoccipital; Pink, Stapes. The directional arrows indicate the orientation of the specimen in each view. Abbreviations: ant, anterior; dor, dorsal; lat, lateral; med, medial; post, posterior; vent, ventral. Scale bars equal 5 cm.

Figure 12 Left laterosphenoid, the supraoccipital, and the basioccipital of NCSM 15728 derived from CT scans.

(A) left laterosphenoid in medial view; (B) left laterosphenoid in lateral view; (C) left laterosphenoid in ventral view; (D) left laterosphenoid in anterior view; (E) left supraoccipital in right lateral view; (F) left supraoccipital in posterior view; (G) basioccipital in ventral view; (H) basioccipital in left lateral view; (I) basioccipital in dorsal view. The directional arrows indicate the orientation of the specimen in each view. Abbreviations: ant, anterior; afb, arched floor of braincase; bf, basioccipital foramen; bk, basioccipital keel; bt, basal tubera; cn, cranial nerve; d, damage; dor, dorsal; hl, head of laterosphenoid; lat, lateral; med, medial; ob, orbitosphenoid boss on laterosphenoid; oc, occipital condyle; pa, parietal articulation; vcms, groove for the vena cerebralis media secunda; vlg, ventral laterosphenoid groove; post, posterior; vent, ventral. Scale bars equal 1 cm.

Figure 13 Left fused opisthotic/exoccipital, left prootic, and the fused basisphenoid/parasphenoid of NCSM 15728 derived from CT scans.

(A) left fused opisthotic/exoccipital in posterior view; (B) left fused opisthotic/exoccipital in anterior view; (C) left fused opisthotic/exoccipital in lateral view; (D) left fused opisthotic/exoccipital in medial view; (E) left prootic in lateral view; (F) left prootic in medial view; (G) fused basisphenoid/parasphenoid in dorsal view; (H) fused basisphenoid/parasphenoid in ventral view; (I) fused basisphenoid/parasphenoid in left lateral view; (J) fused basisphenoid/parasphenoid in posterior view; (K) fused basisphenoid/parasphenoid in anterior view. The directional arrows indicate the orientation of the specimen in each view. Abbreviations: alp, anterolateral processes of basisphenoid; bpp, basipterygoid process; bpro, boss for articulation with proatlas; ci, crista interfenestralis; cn, cranial nerve; cpr, crista prootica; ct, crista tuberalis; cup, cutriform process; dor, dorsal; fm, foramen metoticum; fo, fenestra ovalis; fs, fossa subarcuata; lat, lateral; med, medial; oc, occipital condyle; pop, paroccipital process; post, posterior; prp, preotic pendant; sel, sella turcica; vcd, groove for the vena capitis dorsalis; vcms, groove for the vena cerebralis media secunda; ve, vestibule; vent, ventral. Scale bars equal 1 cm.

Figure 14 The predentary and dentary of NCSM 15728.

(A) predentary in right lateral view; (B) predentary in ventral view; (C) left dentary in lateral view; (D) left dentary in ventral view. The directional arrows indicate the orientation of the specimen in each view. Abbreviations: amf, anterior maxillary fossa; an, angular; ant, anterior; co, coronoid; cp, coronoid process; de, dentary; dor, dorsal; et, ectopterygoid; ju, jugal; lat, lateral; lfpd, lateral foramen of predentary; lg, lateral groove of predentary; med, medial; mx, maxilla; pd, predentary; plp, posterolateral process of premaxilla; plpd, posterolateral process of predentary; post, posterior; sf, surangular foramen; sl, splenial; su, surangular; vppd, ventral process of the predentary. Scale bars in (A) and (B) equal 1 cm. Scale bars in (C) and (D) equal 5 cm.

Figure 15 Posterior jaw elements of NCSM 15728 derived in part from CT scans.

(A) photograph of the right post-dentary jaw elements in natural position; (B) left angular in lateral view; (C) left angular in medial view; (D) left angular in ventral view; (E) left surangular in lateral view; (F) left surangular in medial view; (G) left surangular in dorsal view; (H) left surangular in ventral view. The directional arrows indicate the orientation of the specimen in each view. Abbreviations: aa, articulation surface for angular; an, angular; ant, anterior; bo, basioccipital; ca, articulation surface for coronoid; cp, coronoid process; da, articulation surface for dentary; de, dentary; dor, dorsal; ju, jugal; lat, lateral; lds, lateral depression of surangular; lpf, lateral process foramen; lps, lateral process of surangular; med, medial; mps, medial process of surangular; paa, prearticular articulation surface; post, posterior; qj, quadratojugal; qu, quadrate; rp, retroarticular process; sas, splenial articulations surface; sf, surangular foramen; su, surangular; vent, ventral. Scale bar in (A) equals 5 cm. Scale bars in (B) through (H) equal 1 cm.

Figure 16 Additional figures of left posterior jaw elements of NCSM 15728 derived from CT scans.

(A) left post-dentary elements in medial view; (B) left post-dentary elements in dorsal view; (C) left post-dentary elements in ventral view; (D) left splenial in medial view; (E) left splenial in lateral view; (F) left prearticular in medial view; (G) left prearticular in lateral view; (H) left coronoid in medial view; (I) left coronoid in lateral view; (J) left articular in medial view; (K) left articular in dorsal view. Key to colors: Red, Coronoid; Orange, Surangular; Yellow, angular; Blue, Splenial; Green, Prearticular; Purple, articular. The directional arrows indicate the orientation of the specimen in each view. Abbreviations: aa, articulation surface for angular; ant, anterior; lat, lateral; lpp, lateral process of prearticular; med, medial; post, posterior; ra, ridge for articulation with angular; spa, splenial articulation; vent, ventral. Scale bars in (A) through (C) equal 5 cm. Scale bars in (D) through (K) equal 1 cm.

Figure 17 Supraorbital, accessory supraorbital, and ceratobranchial of NCSM 15728.

(A) left supraorbital and accessory supraorbital in dorsal and slightly medial view; (B) left supraorbital and accessory supraorbital in lateral and slightly dorsal view; (C) anterior articulation facet of left supraorbital in proximal view; (D) left ceratobranchial in medial view; (E) right ceratobranchial in lateral view. The directional arrows indicate the orientation of the specimen in each view. Abbreviations: ant, anterior; asor, accessory supraorbital; dor, dorsal; med, medial; post, posterior; sor, supraorbital; vent, ventral. Scale bars equal 1 cm.

Figure 18 Premaxillary and maxillary dentition of NCSM 15728.

(A) right premaxillary dentition in lateral view; (B) anterior portion of left maxillary dentition in ventrolateral view; (C) posterior portion of left maxillary dentition in ventrolateral view. The directional arrows indicate the orientation of the specimen in each view. Abbreviations: ant, anterior; de, dentary; dor, dorsal; dt, dentary tooth/teeth; mt, maxillary tooth/teeth; mx, maxilla; pm, premaxilla; pmt, premaxillary tooth/teeth; post, posterior; sed, sediment. Scale bars equal 1 cm.

Figure 19 Anterior portion of the left dentary dentition from NCSM 15728.

The directional arrows indicate the orientation of the specimen. Abbreviations: de, dentary; dor, dorsal; dt, dentary tooth/teeth; mt, maxillary tooth/teeth; mx, maxilla; post, posterior; sed, sediment. Scale bar equals 1 cm.

Basis of referrals

TLAM.BA.2014.027.0001 displays all six synapomorphies of Thescelosaurus outlined above, as well as the ‘Y-shaped’ excavation in the dorsal edge of the opisthotic discussed above. This specimen does not possess a supraoccipital foramen and possesses a calcaneum that is included in the midtarsal joint, distinguishing TLAM.BA.2014.027.0001 from T. assiniboiensis and T. garbanii, respectively. The morphology of the frontal, postorbital, and squamosal in this specimen match that reported for the paratype of Thescelosaurus neglectus (Boyd et al., 2009), except that TLAM.BA.2014.027.0001 lacks the extreme rugosities present along the orbital margin of the postorbital in the paratype. The morphology of these elements is significantly different in Thescelosaurus assiniboiensis (see Brown, Boyd & Russell (2011) and description below for details).

NCSM 15728 also displays all six synapomorphies of Thescelosaurus and the two putative synapomophies discussed above. NCSM 15728 lacks the supraoccipital foramen that diagnoses Thescelosaurus assiniboiensis (Brown, Boyd & Russell, 2011). NCSM 15728 cannot be directly compared to the fragmentary holotype of Thescelosaurus garbanii because NCSM 15728 does not preserve any of the tarsal morphologies that are diagnostic of T. garbanii. However, the morphology of NCSM 15728 closely matches that of both the type series of T. neglectus (Gilmore, 1915) and TLAM.BA.2014.027.0001, allowing the former specimen to be indirectly compared to and distinguished from T. garbanii.

Emended diagnosis of Thescelosaurus neglectus

Thescelosaurus neglectus differs from all other basal ornithischian taxa as follows: presence of a groove on the medial surface of the prootic extending from the anterodorsal corner of the trigeminal foramen anteriorly to a foramen that passes between the prootic and the laterosphenoid (NCSM 15728). This species differs from Thescelosaurus garbanii as follows: (1) calcaneum not excluded from the midtarsal joint by the astragalus (USNM 7757; TLAM.BA.2014.027.0001). This species differs from Thescelosaurus assiniboiensis as follows: (1) posterior surface of the squamosal concave dorsoventrally and mediolaterally (convex in T. assiniboiensis: USNM 7758; NCSM 15728; TLAM.BA.2014.027.0001); (2) lack of anteroposteriorly oriented ridges on the articular surface for the postorbital on the squamosal (present in T. assiniboiensis: USNM 7758; NCSM 15728; TLAM.BA.2014.027.0001); (3) presence of a groove on the pterygoid extending from the lateral ridge on the quadrate process onto the mandibular process (absent in T. assiniboiensis: NCSM 15728); (4) absence of a foramen extending from the roof of the braincase through to the dorsal surface of the supraoccipital (autapomorphy of T. assiniboiensis; (Brown, Boyd & Russell, 2011): NCSM 15728; TLAM.BA.2014.027.0001); (5) less than thirty percent of the dorsal surface of the basioccipital contributes to the ventral margin of the foramen magnum (at least one-third in T. assiniboiensis: NCSM 15728; TLAM.BA.2014.027.0001); (6) anterior end of basioccipital ‘V-shaped’ and inserts into the posterior end of the basisphenoid (anterior surface of basioccipital flattened in T. assiniboiensis: NCSM 15728; TLAM.BA.2014.027.0001); and, (7) trigeminal foramen completely enclosed within the prootic (spans between prootic and laterosphenoid in T. assiniboiensis: NCSM 15728).

Several other morphological characters noted on the cranium of NCSM 15728 are apomorphic with respect to all other basal ornithischian taxa. However, owing to the lack of comparative data for T. assiniboiensis and T. garbanii, it cannot be determined if these characters represent autapomorphies of T. neglectus, synapomorphies of the taxon Thescelosaurus, or synapomorphies of a subset of the species referred to Thescelosaurus. These characters are: (1) lack of contact between the ventral process of the lacrimal and the anterodorsal process of the palatine (NCSM 15728); (2) presence of numerous foramina and associated grooves on the dorsal and lateral surfaces of the nasal (NCSM 15728; TLAM.BA.2014.027.0001); and, (3) presence of a groove in the anterior margin of the quadratojugal into which the posteroventral projection of the jugal inserted, causing the anteroventral corner of the quadratojugal to overlap the lateral surface of the posteroventral corner of the jugal (NCSM 15728 and TLAM.BA.2014.027.0001).

Description of the Skull of Thescelosaurus neglectus

The skull of NCSM15728 is well preserved, with portions of every cranial bone represented. Only one bone, the left quadratojugal, is fragmentary (Figs. 1–4). The bones on the right side of the skull remain in their original positions, and the right lower jaw remains in close contact (Fig. 1). Alternatively, many of the bones on the left side of the skull are slightly displaced, including the left frontal, lacrimal, prefrontal, postorbital, squamosal, and jugal (Figs. 2 and 3) in addition to the quadrate, which was removed. The posterior bones of the left lower jaw also are slightly displaced from their original positions. The bones of the palate are slightly displaced, but remain in relative close proximity to their presumed original positions. Many of the bones of the braincase are shifted anteriorly and medially from their original positions (Fig. 4), preventing the construction of an accurate endocast, though the endocast and inner ear of Thescelosaurus was described previously in detail by Galton (1989), and the morphology of this specimen differs only in minor details from that original description.

After my initial observations of the skull of NCSM 15728, the premaxillae were damaged in an apparent attempt to remove the skull from its display by a visitor at NCSM. As a result, the figures presented herein and the CT scans obtained before the damage differ slightly from the current morphology of the skull. Specifically, slight damage occurred to the anteroventral projection of the premaxillae and possibly to other portions of the anterior-most parts of the premaxillae.

The skull of TLAM.BA.2014.027.0001 is less complete and experienced more crushing/damage than that of NCSM 15728. Thus, most of the discussion of the cranial anatomy of T. neglectus that follows is based on NCSM 15728. When observations are based solely on examination of TLAM.BA.2014.027.0001, this is noted in the text. Additionally, any differences noted between NCSM 15728 and TLAM.BA.2014.027.0001 are discussed and interpreted as individual variation within T. neglectus.

Cranium

Premaxilla

The anterior-most portions of the premaxillae are fused. Posterior to the anterior-most edentulous region, the open suture between the premaxillae can be traced on the CT scans throughout their length. The presence of at least partial fusion of the premaxillae is reported in Changchunsaurus, Oryctodromeus, and Zephyrosaurus (Sues, 1980; Varricchio, Martin & Katsura, 2007; Jin et al., 2010). The anterior end of the premaxilla is broadly rounded in lateral view (Fig. 5A). A prominent, posteroventrally concave, ventral projection is present along the midline of the anteroventral tip of the premaxilla. The anterodorsal margin of the premaxilla bears a mediolaterally expanded shelf that increases in transverse breadth posteriorly (Figs. 5A and 5B: ads). The anterodorsal shelf ends just anterior to the contact with the nasals, and the posterolateral corners of the shelf formed prominent projections (damaged on left side), giving the anterodorsal shelf a ‘V-shaped’ outline in dorsal view (Fig. 5B). The dorsal surface of the shelf and the anterior tip of the premaxillae are rugose and covered with foramina (Fig. 5B), as seen in the basal ornithischian Lesothosaurus (Sereno, 1991) and the neornithischians Changchunsaurus, Hypsilophodon, Jeholosaurus, Oryctodromeus, and Zephyrosaurus (Galton, 1974a; Sues, 1980; Varricchio, Martin & Katsura, 2007; Barrett & Han, 2009; Jin et al., 2010). This rugose region likely supported a rhamphotheca (Sereno, 1991).

The posterodorsal processes of the premaxillae arise posterior to the anterodorsal shelf, dividing the anterior processes of the nasal and overlapping their dorsal surfaces (Fig. 5B: pdp). The posterodorsal processes extend along the dorsal surface of the premaxillae farther than in any other neornithischian taxon (Norman et al., 2004), eventually terminating level with the posterior-most extent of the oral margin of the premaxillae (Fig. 5B). The oral margin of the premaxilla is longer than the oral margin of the predentary (Figs. 1 and 2), as seen in the heterodontosaurid Heterodontosaurus (Crompton & Charig, 1962; Norman et al., 2011; Sereno, 2012), and the neornithischian Haya (Makovicky et al., 2011). The lateral surface of the oral margin of the premaxilla is everted (Fig. 5B) as in the neornithischians Agilisaurus, Changchunsaurus, Orodromeus, Oryctodromeus, and Talenkauen (Peng, 1992; Scheetz, 1999; Novas, Cambiaso & Ambrosio, 2004; Varricchio, Martin & Katsura, 2007; Jin et al., 2010) and the basal iguanodontians Dryosaurus, Dysalotosaurus, and Tenontosaurus (Norman, 2004), which results in the premaxillary tooth row being positioned lateral to the maxillary tooth row. The oral margin of the premaxilla is smooth, in contrast to the denticulate oral margin present in basal ankylopollexians (Norman, 2004), and is situated level with the maxillary tooth row (Fig. 5A) and not ventrally deflected as seen in heterodontosaurids (Butler, 2005; Norman et al., 2011; Sereno, 2012), the neornithischians Hypsilophodon and Orodromeus (Galton, 1974a; Scheetz, 1999), and the basal iguanodontian Zalmoxes (Weishampel et al., 2003). There is a short edentulous region anterior to the premaxillary teeth (Fig. 5A), as in all ornithischians (Butler, Upchurch & Norman, 2008), and a diastema is present between the premaxillary and maxillary tooth rows (Fig. 5A), as in all neornithischian taxa except Agilisaurus (Peng, 1992; Barrett, Butler & Knoll, 2005). Six premaxillary teeth are present in each premaxilla, a condition also present in the basal ornithischian Lesothosaurus (Sereno, 1991), the basal thyreophoran Scutellosaurus (Colbert, 1981), and the neornithischian Jeholosaurus (Barrett & Han, 2009). In the lateral surface of the premaxilla, ventral to the rugose anterodorsal shelf, a premaxillary foramen (sensu Sereno, 1991) and a rostral premaxillary foramen (sensu Sereno, 1991) are present, with the former situated directly posterior to the latter (Fig. 5A: pmf and apmf, respectively). Premaxillary foramina also are present in the basal ornithischian Lesothosaurus (Sereno, 1991), the neornithischians Changchunsaurus, Haya, Hypsilophodon, Jeholosaurus, Oryctodromeus, and Zephyrosaurus (Galton, 1974a; Sues, 1980; Varricchio, Martin & Katsura, 2007; Barrett & Han, 2009; Jin et al., 2010; Makovicky et al., 2011), and the basal iguanodontian Zalmoxes (Weishampel et al., 2003). The surface of the premaxilla ventral to the anterodorsal shelf and anterior to the nares is dorsoventrally concave, though a distinct subnarial fossa is not present.

The posterolateral process arises just anterior to the posterior end of the premaxilla, and first angles posterodorsally before curving directly posteriorly, with its ventral margin roughly following the contact between the maxilla and the nasals (Fig. 5A). In NCSM 15728, a small, anterodorsal projection, the premaxillary narial process, is present at the anterodorsal corner of the posterolateral process on both sides of the skull. It wraps around the posterior edge of the external nares (Figs. 5A and 5D: pnp). That feature is not present in any other neornithischian taxon, but it is also absent in TLAM.BA.2014.027.0001, suggesting this feature is either unique to NCSM 15728 or is polymorphic within T. neglectus. The posterolateral process of the premaxilla does not extend far enough posteriorly to contact the lacrimal (Fig. 1), unlike in the heterodontosaurid Heterodontosaurus (Norman et al., 2004; Norman et al., 2011; Sereno, 2012), the neornithischian Jeholosaurus (Barrett & Han, 2009), the basal ceratopsians Liaoceratops and Yinlong (You & Dodson, 2003; Xu et al., 2006), and most basal iguanodontians (e.g., Tenontosaurus; Norman, 2004). The posterolateral process is not as dorsoventrally tall as in Parksosaurus (Galton, 1973).

The palatal surface of the premaxillae is concave anteriorly (Fig. 5E). At the level of the second tooth position a ridge is present along the midline of the premaxillae, extending to the posterior end of the premaxillae. Based on examination of the CT data and the presence of slight transverse crushing in this specimen, the ridge is likely a taphonomic feature. The majority of the palatal surface was flat. A pair of rostral palatal foramina (sensu Sereno, 1991) are present anterior to the first premaxillary tooth (Fig. 5E: rpf). Similar foramina are present in the basal ornithischian Lesothosaurus (Sereno, 1991), the neornithischians Changchunsaurus and Zephyrosaurus (Sues, 1980; Jin et al., 2010), and some marginocephalians (e.g., Archaeoceratops; (You & Dodson, 2003)). The rostral palatal foramina connect to the rostral premaxillary foramina, as suggested previously by several authors (e.g., Sereno, 1991; Jin et al., 2010). The slit-like opening present along the midline of the palatal surface seen in Changchunsaurus is absent in NCSM 15728 (Jin et al., 2010). In the ventrolateral corner of the posterior end of the premaxilla a concavity is present. The concavity receives the short anterolateral process of the maxilla (Fig. 5C: pls), as in Changchunsaurus, Haya, Orodromeus, Oryctodromeus, and Zephyrosaurus (Sues, 1980; Scheetz, 1999; Varricchio, Martin & Katsura, 2007; Jin et al., 2010; Makovicky et al., 2011). Posteromedially, the anterior processes of the maxillae meet along the midline and insert into the posterior end of the premaxilla dorsal to the palatal shelf. The anterior end of the vomer is positioned ventral to the anterior-most end of the maxilla and its anterior tip inserts into a shallow concavity in the posteromedial end of the premaxillae ventral to the paired maxillae (Fig. 5C).

Nasal

The nasal is an anteroposteriorly long element that is strongly concave ventromedially, equal in length to the frontal, and thin throughout its length. The nasals meet along the midline, but transverse compression of the specimen caused the nasals to crush together slightly, obscuring the original morphology of their contact. There is no evidence of a midline depression on the nasals (Fig. 3) as seen in the heterodontosaurid Heterodontosaurus, the neornithischians Agilisaurus, Changchunsaurus, Haya, Hexinlusaurus, Jeholosaurus, and the basal ceratopsian Yinlong (Jin et al., 2010; Makovicky et al., 2011). The anterior end of the element was sharply pointed and its anterolateral margin formed the posterodorsal corner of the external nares (Figs. 1 and 2). The anterior tips of the nasals were separated by the posterodorsal processes of the premaxillae (Figs. 3 and 5B), which inserted between the nasals anteriorly and then transitioned to overlapping the nasals at their posterior ends. The nasals are also divided anteriorly by the posterodorsal processes of the premaxillae in Hypsilophodon, but this condition is absent in other neornithischian taxa (e.g., Haya and Jeholosaurus: Barrett & Han, 2009; Makovicky et al., 2011).

The lateral edge of the nasal is curved ventrally and overlapped the lacrimal and maxilla laterally (Figs. 1 and 2). The posterolateral corner of the nasal forms part of the dorsal margin of the antorbital fenestra (Figs. 1 and 2). The posterolateral process of the premaxilla overlapped the anterior half of the ventrolateral margin of the nasal, but this contact did not extend all the way to the lacrimal as in the heterodontosaurid Heterodontosaurus (Crompton & Charig, 1962), the neornithischian Jeholosaurus (Barrett & Han, 2009), and the basal ceratopsians Liaoceratops and Yinlong (Xu et al., 2002; Xu et al., 2006). The posterior ends of the nasal were separated by the anterior processes of the frontals and overlapped posterolaterally by the prefrontals. These contacts resulted in the exposure of only a small, tapering wedge of the posterior end of the nasal in dorsal view (Fig. 3). A series of foramina pierce the dorsal and lateral surfaces of the nasal in the area between the posterior-most extent of the posterodorsal processes of the premaxillae and the anterior-most extent of the prefrontals (Figs. 1–3). Shallow grooves extend from some of these foramina onto the surface of the nasal, and examination of the CT images shows that many of these foramina are interconnected and exit the medial surface of the nasal. Their positions and number vary on each side of the skull. In Jeholosaurus, a row of three foramina are present along the ventrolateral margin of the nasals (Barrett & Han, 2009). By contrast, a single foramen is present on the surface of the nasal in Haya (Makovicky et al., 2011). No foramina are reported on the nasal in Hypsilophodon (Galton, 1974a) and none are observed in the preserved portion of the nasal in the holotype of Parksosaurus (Galton, 1973; C Boyd, pers. obs., 2011).

Prefrontal

The prefrontal is a triradiate bone that forms the anterodorsal corner of the orbit and is exposed on the dorsal and lateral surfaces of the skull (Figs. 1 and 2). In lateral view the prefrontal is triangular, with the posterior portion dorsoventrally thicker than the anterior portion (Fig. 6C). A rugose boss is present on the lateral surface of the prefrontal at its dorsoventrally thickest point, immediately adjacent to the anterodorsal corner of the orbit (Fig. 6C: rso). This boss formed part of the articulation surface for the supraorbital along with an adjacent area on the lacrimal (Figs. 6A and 6C) as in other neornithischians (e.g., Hypsilophodon, Parksosaurus: (Galton, 1973; Galton, 1974a)). The orbital margin of the prefrontal transitions from broadly convex immediately posterior to the supraorbital boss to sharply pointed and slightly rugose posteriorly (Fig. 6C).

The dorsal surface of the prefrontal is anteroposteriorly convex and is pierced by a foramen along the dorsomedial margin of the supraorbital boss (Fig. 6C), a condition that is unique to Thescelosaurus (Boyd et al., 2009). This foramen passes ventrolaterally through the prefrontal, exiting into anterodorsal corner of the orbit just ventral to the supraorbital boss. The anterior process of the prefrontal is dorsoventrally thin, ventromedially concave, and rests in a shallow fossa on the dorsal surface of the nasal. The pointed, triangular tip of this process is positioned dorsal to the lacrimal and is bordered anteriorly by the nasal, a condition seen in most basal ornithischians (Norman et al., 2004; Norman, Witmer & Weishampel, 2004a), but not in Parksosaurus wherein the anterior tip inserts between the lacrimal and the dorsal process of the maxilla, nearly preventing the anterior process of the lacrimal from contacting the dorsal process of the maxilla (Galton, 1973; C Boyd, pers. obs., 2011). The posterior process of the prefrontal is dorsoventrally thicker than the anterior process (Fig. 6C). The posterior process wraps around the dorsolateral corner of the anterior end of the frontal while only overlapping the dorsal surface at its posterior-most extent. The ventral process of the prefrontal is not exposed on the exterior of the skull. It arises ventral and slightly posterior to the supraorbital boss and extends ventromedially. The distal end of the ventral process is flattened to slightly concave to fit against a facet on the dorsomedial edge of the lacrimal.

Lacrimal

The lacrimal forms much of the anterior margin of the orbit and the posterodorsal corner of the external antorbital fenestra (Figs. 1, 2 and 6A). It is composed of posteroventral and anterior processes oriented at an angle of approximately 100° (Fig. 6A). The lateral surface of the posteroventral process is dorsoventrally concave and anteroposteriorly convex. The distal end of the posteroventral process is positioned posterior to the maxilla, and dorsal to the anterior tip of the jugal (Fig. 6A), a condition also seen in Orodromeus (Scheetz, 1999). Alternatively, in Gasparinisaura and Jeholosaurus the posteroventral tip of the lacrimal is situated anterior to the jugal and posterodorsal to the maxilla Coria & Salgado, 1996; Barrett & Han, 2009 and in Hypsilophodon it is dorsal to both the jugal and the maxilla (Galton, 1974a). The anterior process also did not contact the dorsal process of the maxilla on the lateral surface of the skull (Fig. 6A), unlike the condition seen in the neornithischians Changchunsaurus, Haya, and Parksosaurus (Galton, 1973; Jin et al., 2010; Makovicky et al., 2011).

The foramen for the prominent lacrimal duct is present on the dorsal portion of the posterior surface of lacrimal (Fig. 6B). This foramen penetrates the middle of the anterior process and eventually opens along the medial surface near the distal end of the anterior process. The posterodorsal corner of the lateral surface of the lacrimal is rugose where it contacted the base of the supraorbital (Fig. 6A: aso). Anteroventral to this rugose area, foramina pierce the lateral surface of the lacrimal. On the right side there are two foramina, while on the left there are three. The posterodorsal margin of the lacrimal contacts the prefrontal. The nasal overlaps much of the dorsal and lateral surfaces of the anterior process of the lacrimal, preventing the anterior process from contacting the posterolateral process of the premaxilla. Contact between the lacrimal and the premaxilla is present in Heterodontosaurus (Crompton & Charig, 1962), Jeholosaurus (Barrett & Han, 2009), some basal ceratopsians (e.g., Liaoceratops and Yinlong: Xu et al., 2002; Xu et al., 2006), and some basal iguanodontians (e.g., Tenontosaurus, Dryosaurus; Norman, 2004). The ventrolateral margin of the anterior process projects ventrally as a mediolaterally thin sheet over the posterodorsal corner of the antorbital fossa. A mediolaterally thin sheet of bone extended from the anteromedial margin of the posteroventral process across to the ventromedial margin of the anterior process, forming the posterodorsal portion of the medial wall of the antorbital fossa. The ventral margin of this sheet is slightly thickened and contacted a corresponding medial sheet of the maxilla (Figs. 6A and 6B: drmm). The medial surface of the ventral process did not contact the palatine, unlike in Hypsilophodon, Jeholosaurus, and Lesothosaurus (Galton, 1974a; Sereno, 1991; Barrett & Han, 2009).

Maxilla

The maxilla forms the anterior and ventral margins of the antorbital fenestra, but is excluded from bordering the external nares anteriorly by the posterolateral process of the premaxilla (Figs. 1 and 2). The maxillary tooth row is shorter than the dentary tooth row (Figs. 1 and 2). There is a shallow fossa present on the anteroventral corner of the lateral surface of the maxilla, just posterior to the contact with the premaxilla (Figs. 1, 2 and 5A). This fossa is also present in Changchunsaurus, Haya, Hypsilophodon, Jeholosaurus, Orodromeus, and Zephyrosaurus (Butler, Upchurch & Norman, 2008; Jin et al., 2010; Makovicky et al., 2011). There are twenty tooth positions in the maxilla of NCSM 15728, but only eighteen in TLAM.BA.2014.027.0001. This discrepancy is either a result of individual variation, or perhaps an ontogenetic difference because the latter specimen is slightly smaller than NCSM 15728 (Table 1). The maxillary tooth row ends level with the posterior edge of this lateral maxillary fossa, creating a flat diastema between the maxillary and premaxillary tooth rows. In heterodontosaurids, the maxillary diastema is anteroposteriorly concave (Butler, Upchurch & Norman, 2008). Just anterior to the lateral maxillary fossa a short, anterolateral boss is present that inserted into a posterolateral recess in the premaxilla (Fig. 5C: almp), a character shared by Changchunsaurus, Haya, Orodromeus, Oryctodromeus (inferred based on the morphology of the premaxillae), and Zephyrosaurus (Sues, 1980; Scheetz, 1999; Jin et al., 2010; Makovicky et al., 2011; C Boyd, pers. obs., 2011). This boss is separate from the long, ‘spike-like’ process that forms the anterior-most end of the maxilla and inserts deeply into the posterior end of the premaxilla (Scheetz, 1999). The anterior ends of the maxillae contact each other medially, after inserting into the premaxillae. Where the maxillae are in contact medially, the vomer overlaps their ventral surfaces until the maxillae insert into the posterior end of the premaxillae, though posterior to this contact the vomer inserts between the medial surfaces of the maxillae.

The lateral surface of the maxilla is overlapped dorsally by the nasal and anteriorly by the posterolateral process of the premaxilla (Figs. 1 and 2). A small, dorsally directed, triangular projection is positioned ventral to the nasal and formed the anterior boarder of the antorbital fenestra. A prominent anteroposteriorly oriented ridge is present on the lateral surface of the maxilla, causing the tooth row to be inset medially. In Lesothosaurus and Scutellosaurus this ridge is reduced in size, resulting in only a slight emargination (Colbert, 1981; Sereno, 1991). A few small foramina pierce the surface of this ridge near its apex, and a row of larger foramina are present ventral to this ridge. The maxillary border of the external antorbital fenestra is anteroposteriorly concave and is sharply defined along its entire length, unlike in the heterodontosaurid Abrictosaurus (Thulborn, 1974), the thyreophorans Emausaurus and Scelidosaurus (Butler, Upchurch & Norman, 2008), the basal ornithischian Lesothosaurus (Sereno, 1991), the neornithischian Zephyrosaurus (Sues, 1980), and the basal ceratopsian Archaeoceratops (You & Dodson, 2003) where the external antorbital fenestra rounds smoothly on the maxilla along at least a portion of its margin. Unlike in the neornithischians Haya and Hypsilophodon (Galton, 1974a; Makovicky et al., 2011), there is no maxillary fenestra present anterior to the antorbital fenestra. The posterodorsal margin of the maxilla contacts the lacrimal and jugal along a continuous butt joint (Figs. 1 and 2).

The medial surface of the maxilla is dorsoventrally concave. Near the ventral margin a row of replacement foramina are present dorsomedial to the tooth row, as in all neornithischians and the heterodontosaurid Fruitadens (Norman et al., 2004; Butler et al., 2010). Just anterior to the external antorbital fenestra a mediolaterally thin medial process extends dorsally. Anteriorly, this medial process extends dorsally and connects to the dorsomedial surface of the triangular projection of the maxilla anterior to the antorbital fenestra, creating a small internal antorbital fenestra in the anteroventral corner of the antorbital fossa (Figs. 2B and 6B: iaof). This medial process extends posteriorly, forming the medial and much of the dorsal walls of the antorbital fossa. Posteriorly, the medial process contacts a medial sheet of bone extending from the lacrimal and gradually reduces in dorsoventral height until it reaches the contact between the maxilla and the ventral process of the lacrimal. The dorsal margin of the medial process of the maxilla is mediolaterally expanded where it contacts the lacrimal (Fig. 6A: drmm). A small fenestra is also present in the posteroventral corner of the antorbital fenestra, between the maxilla and the lacrimal, that opened posteriorly into the orbit.

Jugal

The jugal forms the entire ventral, and part of the anterior, margin of the infratemporal fenestra as well as the entire ventral, and part of the posterior, margin of the orbit (Fig. 1). The lateral surface of the jugal lacks the ornamentation seen in Jeholosaurus (Barrett & Han, 2009) and either a low (Changchunsaurus: Jin et al., 2010) or pronounced jugal boss (Orodromeus, Zephyrosaurus, and an unnamed taxon from the Kaiparowits Formation of Utah: Sues, 1980; Scheetz, 1999; Boyd, 2012; Gates et al., 2013). The anterior process of the jugal is straight in lateral view (Fig. 1), unlike the curved anterior process seen in the neornithischians Agilisaurus and Zephyrosaurus (Peng, 1992; Scheetz, 1999). It is dorsoventrally deeper than mediolaterally broad, unlike in thyreophorans (Norman, Witmer & Weishampel, 2004b). The anterior process of the jugal is excluded from contacting the margin of the antorbital fenestra by the lacrimal and the maxilla, as in all non-cerapodan neornithischians (Norman et al., 2004), Hypsilophodon (Galton, 1974a), and many basal iguanodontians (e.g., Gasparinisaura, Zalmoxes: Coria & Salgado, 1996; Weishampel et al., 2003). The tip of the anterior process is triangular in shape, and ends dorsal to the maxilla (Fig. 1), in contrast to the neornithischians Agilisaurus and Hypsilophodon (Galton, 1974a; Peng, 1992) and most iguanodontians (e.g., Dysalotosaurus, Gasparinisaura, and Tenontosaurus: Coria & Salgado, 1996; Norman, 2004) where the anterior process of the jugal inserts into the maxilla. The dorsal surface of the tip of the anterior process of the jugal forms an extensive butt-joint against the ventral process of the lacrimal (Fig. 6A).

Medially, the dorsal and ventral margins of the anterior process are thickened, the jugal forms an extensive butt-joint against the ventral process of the lacrimal (Fig. 6A) making the medial surface dorsoventrally concave. On the medial surface of the anterior process, an elongate, anteroposteriorly oriented groove is present that formed the articulation surface for the ectopterygoid (Fig. 7B: mgj), as in all basal ornithischians. The dorsal process of the jugal is the most gracile of the three processes on the jugal and angles posterodorsally to contact the postorbital. The contact surface for the postorbital on the dorsal process of the jugal faces laterally and slightly anteriorly (Fig. 7A: apo).

The medial surface of the dorsal process is concave anteroposteriorly, and its anterior edge is thicker than the posterior edge. The dorsal and posterior processes of the jugal form an oblique angle at the anteroventral corner of the infratemporal fenestra. The dorsoventral height of the posterior process is less than 25% of the total height of the skull, as in the neornithischians Agilisaurus, Haya, Hexinlusaurus, Jeholosaurus, and Orodromeus (He & Cai, 1984; Peng, 1992; Scheetz, 1999; Barrett & Han, 2009; Makovicky et al., 2011) and the basal ceratopsian Yinlong (Xu et al., 2006). The posterior process is bifurcated at its distal end, giving rise to an elongate, ‘tab-shaped’ dorsal projection and a triangular ventral projection. The dorsal projection overlapped the lateral surface of the quadratojugal along the ventral margin of the infratemporal fenestra (Figs. 7A and 7B: dpj), while the ventral projection inserted medial to the quadratojugal (Figs. 7A and 7B: vpj). The lateral surface of the ventral margin of the posterior process is depressed and covered by a series of ridges (Fig. 7A: vd), a feature only known in the taxon Thescelosaurus (Boyd et al., 2009).

Quadratojugal

The quadratojugal is a mediolaterally thin, ‘plate-like’ bone that formed a small part of the posterior margin of the infratemporal fenestra (Fig. 1). A thin, anteroposteriorly flattened projection of bone expands dorsally along the anterior margin of the quadrate, wrapping anteriorly and medially to the dorsal portion of the jugal wing on the quadrate. This dorsal process did not reach the ventral process of the squamosal, unlike in the heterodontosaurid Heterodontosaurus (Crompton & Charig, 1962), the neornithischian Lesothosaurus (Sereno, 1991), and the basal iguanodontians Dryosaurus and Dysalotosaurus (Norman et al., 2004). The dorsal margin of the quadratojugal posterior to the dorsal process is posterodorsally concave to wrap around the anterior margin of the jugal wing of the quadrate. The posterior margin of the quadratojugal is slightly concave with rounded posterodorsal and posteroventral corners. The medial surface of the posteroventral corner of the quadratojugal contacted the quadrate along a laterally flattened facet just dorsal to the distal condyles (Fig. 8D: aqj). The ventral margin of the quadratojugal is sloped anterodorsally.

The anterior portion of the quadratojugal participates in a complicated contact with the posterior process of the jugal. The majority of the anterior end of the quadratojugal inserted medial to the posterior process of the jugal; however, the anteroventral corner of the quadratojugal possesses a dorsoventrally oriented groove that the posterior process of the jugal inserted into, which causes the posteroventral corner of the posterior process of the jugal to insert medial to the quadratojugal (Fig. 1). Thus, the jugal overlaps the lateral surface of the quadratojugal dorsally and inserts medial to the quadratojugal ventrally. This morphology is unique to this specimen, but since the quadratojugal is not preserved in Thescelosaurus assiniboiensis or Thescelosaurus garbanii (Morris, 1976; Brown, Boyd & Russell, 2011), it is uncertain if this morphology is an autapomorphy of Thescelosaurus neglectus or a synapomorphy of Thescelosaurus. A similar condition is seen in the basal iguanodontians Tenontosaurus and Zalmoxes, except that in those taxa the quadratojugal sits in a dorsoventral groove in the jugal, producing the same pattern of overlap on the lateral surface of the skull (Weishampel et al., 2003; Godefroit, Codrea & Weishampel, 2009). A small quadratojugal foramen is present slightly posterior to the contact between the jugal and the quadratojugal (Fig. 1), which is also present in the neornithischians Haya, Hypsilophodon, Jeholosaurus, Parksosaurus, and some specimens of Orodromeus (e.g., MOR 1141) and the basal iguanodontian Tenontosaurus tilletti (Galton, 1973; Galton, 1974a; Scheetz, 1999; Norman, 2004; Barrett & Han, 2009; Makovicky et al., 2011).

Postorbital

The postorbital formed the posterodorsal corner of the orbit, the anterodorsal margin of the infratemporal fenestra, and the anterolateral margin of the supratemporal fenestra (Fig. 1). The postorbital consists of two prominent processes directed ventrally and posteriorly, and a third, reduced process directed anteriorly (Fig. 7C). The ventral process is triangular in transverse section, with the lateral surface anteroposteriorly concave. The ventral process overlaps the lateral surface of the dorsal process of the jugal, as in the neornithischians Agilisaurus, Jeholosaurus, Parksosaurus, and Zephyrosaurus (Galton, 1973; Sues, 1980; Peng, 1992; Barrett & Han, 2009). The short anterior process extends anterior from the contact between the frontal and postorbital and envelopes the lateral and ventral margins of the frontal (Fig. 7C: app). The orbital margin of the main body of the postorbital and the anterior process is rugose as seen in the neornithischians Haya, Orodromeus, and Zephyrosaurus (Sues, 1980; Scheetz, 1999; Makovicky et al., 2011) and the basal ceratopsians Archaeoceratops and Liaoceratops (Xu et al., 2002; You & Dodson, 2003). A distinct anteriorly directed inflation is present along the orbital margin (Fig. 7C: aip), as in the neornithischians Haya, Hexinlusaurus, Jeholosaurus, Orodromeus, Thescelosaurus assiniboiensis, and Zephyrosaurus (Sues, 1980; He & Cai, 1984; Scheetz, 1999; Barrett & Han, 2009; Brown, Boyd & Russell, 2011; Makovicky et al., 2011, C Boyd, pers. obs., 2011). A prominent, anteroposteriorly oriented ridge extends from the dorsal margin of this projection posteriorly along the lateral surface of the postorbital onto the posterior process. Ventral to this ridge the surface of the postorbital is flattened (Fig. 7C: soaa). It was proposed that this anterior projection into the orbit served as a site of attachment for the supraorbital or, when present, the accessory supraorbital (Norman et al., 2004). This hypothesis is confirmed by the fact that the accessory supraorbital in this specimen rests on the flattened lateral surface of this projection ventral to the anteroposteriorly oriented ridge (Fig. 1). Posterior to this contact surface for the accessory supraorbital a series of small foramina are present, though the number and position vary on each side of the specimen.

The posterior process angles posterodorsally and its lateral surface is dorsoventrally concave. The posterior process twists about its long axis so that its lateral surface rotates to face dorsolaterally (Figs. 3 and 7C). The distal end is bifurcated into medial and lateral projections, with the lateral projection extending farther posteriorly (Fig. 8F: mp and lp, respectively). These projections insert into the anterior process of the squamosal, which is also bifurcated into mediolaterally broad dorsal and ventral projections, with the ventral projection extending further anteriorly than the dorsal projection. These four projections tightly interlock with each other, forming a secure contact between these two elements (Fig. 8F). The main body of the postorbital is relatively mediolaterally thin, unlike the robust postorbital seen in some basal iguanodontians (e.g., Tenontosaurus and Zalmoxes: Norman, 2004; Weishampel et al., 2003) and ankylopollexians (e.g., Camptosaurus: Norman, 2004). On the ventromedial surface adjacent to the contact surface for the frontal, a prominent facet is present for the head of the laterosphenoid. This contact surface extends medially onto the frontal.

Frontal

The frontals are dorsally flattened, anteroposteriorly longer than wide, and approximately the same length as the nasals (Fig. 3). Each frontal is roughly triangular in dorsal view, with the anterior end pointed and the posterior end transversely wide. The medial margins of the frontals remain in contact throughout their entire length and the medial contact surface consists of a series of anteroposteriorly oriented ridges and grooves. The anterior tips of the frontals insert in between the posterior ends of the nasals and overlap the dorsal surface of the posteromedial corners of the nasals. The anterolateral portion of the frontal is dorsally depressed, creating a facet into which the posterior process of the prefrontal inserted. The frontals extend over the entire orbit, unlike in Zalmoxes and some ankylopollexians where the frontals are only positioned over the posterior half of the orbit (Weishampel et al., 2003; Norman, 2004). The frontal forms the middle portion of the orbital margin, and is dorsoventrally thin and rugose along this margin as most neornithischian taxa (e.g., Haya, Zephyrosaurus, and an unnamed taxon from the Kaiparowits Formation of Utah: Sues, 1980; Makovicky et al., 2011; Boyd, 2012; Gates et al., 2013). The orbital contribution of the frontal is less than 25% of the total length of the frontal, as in Thescelosaurus assiniboiensis (Brown, Boyd & Russell, 2011) and some basal iguanodontians (e.g., Muttaburrasaurus: Bartholomai & Molnar, 1981). The width of the frontals is greatest at mid-orbit level, not across the posterior end (Fig. 3), a condition unique to Thescelosaurus (Boyd et al., 2009).

The postorbital contacts the posterolateral corner of the frontal, and the articulation facet for the postorbital is oriented laterally and wraps around to the ventral surface, as in Zephyrosaurus, but unlike the dorsally facing articulation facet seen in an unnamed taxon from the Kaiparowits Formation of Utah (Boyd, 2012; Gates et al., 2013). The articulation between the frontal and postorbital consists of a series of pronounced, interlocking projections, as in Hypsilophodon, Orodromeus, and Zephyrosaurus (Galton, 1974a; Scheetz, 1999; C Boyd, pers. obs., 2011). The articulation surface for the dorsal head of the laterosphenoid is positioned on the ventral surface along the contact between the frontal and the postorbital, which is also seen in the neornithischians Agilisaurus, Jeholosaurus, Lesothosaurus, Thescelosaurus assiniboiensis, Zephyrosaurus, and an unnamed taxon from the Kaiparowits Formation of Utah (Sues, 1980; Sereno, 1991; Barrett, Butler & Knoll, 2005; Barrett & Han, 2009; Brown, Boyd & Russell, 2011; Boyd, 2012). The frontals form the anteromedial margins of the supratemporal fenestrae. The posterior-most extent of each frontal is along the midline. These projections inserting into corresponding slots in the anterodorsal surface of the parietal, as in Thescelosaurus assiniboiensis (Brown, Boyd & Russell, 2011). In Hypsilophodon posterior projections are also present, but they are positioned slightly lateral to the midline (Galton, 1974a), and in Haya, Lesothosaurus, and Orodromeus the posterior contact with the parietals is relatively straight (Sereno, 1991; Scheetz, 1999; Makovicky et al., 2011). There is a broad, ventrolaterally oriented concavity on the ventral surface along the orbital margin, the limits of which are denoted by the presence of a sharp, ventrally pointing ridge. Medial to this ridge, the ventromedial surface of the frontal is concave where the olfactory bulb and tract and the anterior portion of the cerebrum were positioned (Galton, 1989). This ventromedial concavity is more pronounced than the ventrolateral concavity. The posterior end of the frontal is dorsoventrally thicker than the anterior end.

Parietal

The parietals are completely fused, and form much of the anterior, medial, and posterior margins of the supratemporal fenestrae (Fig. 3). Anteriorly, the parietals make a mediolaterally broad contact with the posterior margin of the frontals (Fig. 3). The median process (sensu Galton, 1974a) is situated along the midline of the anteroventral margin of the parietals and inserted into a shallow notch in the posteroventral surface of the frontals. Dorsoventrally thin, mediolaterally wide processes extend anteriorly from the anterolateral corners of the parietals. These processes were appressed to the ventral surface of the frontals, slightly overlapped the posteroventral portion of the contact between the postorbital and the frontal, and ended just posterior to the articulation surface on the frontal and postorbital for the laterosphenoid. The ventral surface of these processes formed an anteroposteriorly long, mediolaterally concave contact with the dorsal surface of the laterosphenoid.

The dorsomedial portion of the anterior margin of the parietals is indented to receive the two posteromedial projections of the frontals. Just posterior to these indentations on the dorsal surface of the parietals, a flattened, triangular-shaped surface is present that narrows posteriorly leading to the sagittal crest. The sagittal crest is a narrow ridge with steeply sloped lateral surfaces that extends to the posterior margin of the parietals. The lateral surfaces of the parietals are anteroposteriorly concave and dorsoventrally convex below the sagittal crest, giving the parietals an hourglass shape in dorsal view (Fig. 3). In the middle of the lateral surface, the ventral half is covered by a series of posterodorsally inclined ridges, though matrix obscures exactly how many ridges were present and how far posteriorly they extend. In each of the posterodorsal corners of the lateral surfaces a pronounced fossa is present. In dorsal view, the posterior margin is posteriorly concave, with the lateral wings meeting along the midline at a sharp angle, and a thickened ridge is present along the entire posterior border. The posterolateral surfaces contacted mediolaterally thin, ventromedially directed processes of the squamosals. Ventrally, the parietals are deeply concave for receiving the supraoccipital. At the posterior margin there is a ventrally directed wedge of bone along the midline of the element (Fig. 4). The lateral walls of the parietals are mediolaterally thin posteriorly, with their ventral tips wedged between the squamosal and the supraoccipital. The lateral walls thicken and decrease in dorsoventral height anteriorly, reaching their maximum mediolateral thickness at the posterior margin of their contact with the laterosphenoids.

Squamosal

The squamosal forms the dorsal margin of the infratemporal fenestra and the posterolateral margin of the supratemporal fenestra (Figs. 1, 3 and 8A). It has four distinct processes. The anterior process curves ventrally as it approaches and contacts the postorbital (Fig. 8A). The contact surface for the postorbital consists of two anteriorly directed projections, the ventral projection being longer than the dorsal projection (Fig. 8A: vsq and dsq, respectively). The ventral projection was positioned ventromedial to the posterior process of the postorbital, while the dorsal projection overlapped the dorsolateral surface of the posterior process (Fig. 8F). The posterior process was forked into medial and lateral projections that inserted into anteroposteriorly elongate grooves on either side of the dorsal projection of the squamosal (Fig. 8B). This same contact is present in the paratype of Thescelosaurus neglectus (USNM 7758), but the contact in Thescelosaurus assiniboiensis is not as intricate (Boyd et al., 2009; Brown, Boyd & Russell, 2011). Additionally, the series of anteroposteriorly oriented ridges present on the dorsal surface of the articulation with the postorbital in Thescelosaurus assiniboiensis are absent in NCSM 15728 and TLAM.BA.2014.027.0001 (Boyd et al., 2009). The anteroventrally directed prequadratic process (sensu Makovicky et al., 2011), is triangular in transverse section, arises anterior to the socket for the head of the quadrate, and extends ventrally along the anterior surface of the quadrate with its distal end tapering to a point (Fig. 8A: prq).

The prequadratic process does not extend far enough ventrally to contact the dorsal process of the quadratojugal, unlike in the heterodontosaurid Heterodontosaurus (Crompton & Charig, 1962), the basal ornithischian Lesothosaurus (Sereno, 1991), and the basal iguanodontians Dryosaurus and Dysalotosaurus (Norman, 2004). The posteroventral, or postquadratic (sensu Makovicky et al., 2011) process is an anteroposteriorly thin sheet that forms much of the posterior surface of the squamosal, enclosing the posterior end of the socket for the head of the quadrate (Fig. 8A: poq). The lateral margin of the postquadratic process flares posterolaterally, creating a broad lateral wing (Fig. 8B). The posterodorsal surface of the squamosal is posteromedially concave in dorsal view and in lateral view the posterior margin is offset at a right angle from the posterodorsal margin (Fig. 8A), as in the paratype of Thescelosaurus neglectus (USNM 7758: Boyd et al., 2009). Alternatively, the posterodorsal corner of the squamosal is convex in lateral views and the posterior margin is concave to straight in dorsal view in Thescelosaurus assiniboiensis (Brown, Boyd & Russell, 2011). The convex anterior surface of the paroccipital process fit into the posterior concavity on the squamosal. The medial process of the squamosal is a stout sheet of bone that extends anteromedially from the posteromedial margin of the postquadratic process (Fig. 4).

The medial process narrows in dorsoventral height as it extends medially, and its medial end possesses an anteroposteriorly elongate groove. The posteroventral end of the parietals inserted into this groove, and a small projection of the medial process of the squamosal cupped the ventral surface of the parietal, preventing the latter element from contacting the supraoccipital along is posteroventral surface. The dorsal surface of the squamosal is medially expanded to unite the dorsomedial margin of the anterior process and the anterodorsal margin of the medial process (Fig. 8B), creating a dorsally enclosed pocket in the posterolateral corner of the supratemporal fenestra.

A thin, sharply defined, anteroposteriorly oriented ridge arises on the ventral surface of the anterior process of the squamosal (Fig. 8A: rsq). This ventral ridge extends posteriorly to the base of the prequadratic process of the squamosal, becoming dorsoventrally taller. A ventral ridge also extends between the posterior margin of the prequadratic process and the anterior margin of the postquadratic process, enclosing the medial surface of the socket for the head of the quadrate. In ventral view, these ventral ridges divide the ventral surface of the squamosal into lateral and medial fossae. The medial fossa is twice the transverse breadth of the lateral fossa on average. Anterior to the prequadratic process the lateral fossa forms a well-developed, ventrolaterally oriented concavity (Fig. 8A) for the adductor musculature (M. adductor mandibulae superficialis; Galton, 1974a; Jin et al., 2010). A smooth, laterally facing surface extends posteriorly from this concavity dorsal to the socket for the head of the quadrate, reaching the anterior margin of the postquadratic process. There is no parietosquamosal shelf, unlike the condition in all known marginocephalian taxa (Butler, Upchurch & Norman, 2008).

Palatoquadrate

The palatoquadrate region of NCSM 15728 is relatively well preserved, though many of the elements have been slightly displaced from their natural positions. Elements from the left side of the skull of NCSM 15728 are figured, but their morphology is congruent with that of their antimeres. The isolated right quadrate preserved with TLAM.BA.2014.027.0001 is used to describe portions of the quadrate that were difficult to discern from NCSM 15728.

Quadrate

The quadrate shaft leans posteriorly in lateral view (Fig. 1). The ventral portion of the quadrate shaft angles slightly anteroventrally, unlike in some basal ceratopsians (e.g., Yinlong) where the shaft angles posteroventrally (Xu et al., 2006). The distal condyles of the quadrate are dorsolaterally sloped in posterior view, as seen in the neornithischians Jeholosaurus, Orodromeus, Oryctodromeus, and Zephyrosaurus (Sues, 1980; Scheetz, 1999; Barrett & Han, 2009; C Boyd, pers. obs., 2011). The quadrate was separated from the jugal by the quadratojugal, unlike in some basal iguanodontians and ankylopollexians (e.g., Dryosaurus, Camptosaurus: Norman, 2004). The contact surface for the quadratojugal begins ventrally on the lateral surface of the quadrate, just dorsal to the distal condyles (Fig. 8D: aqj), which is the basal ornithischian condition. The contact extends dorsally along the lateral surface of the ventral third of the quadrate shaft, then wraps around to the anteromedial surface where the dorsal process of the quadratojugal contacts the quadrate shaft. Overall, the contact between the quadrate and the quadratojugal extends along more than half of the dorsoventral height of the quadrate, unlike the reduced contact seen in the neornithischian Changchunsaurus (Jin et al., 2010), the basal ceratopsians Archaeoceratops and Yinlong (You & Dodson, 2003; Xu et al., 2006), and the basal iguanodontians Dryosaurus and Dysalotosaurus (Norman, 2004). A foramen is present in the lateral surface of the quadrate, just posterior to the contact with the quadratojugal (Figs. 8D and 8F: qf). A similar foramen is present in the neornithischians Haya and Parksosaurus (Makovicky et al., 2011; C Boyd, pers. obs., 2011) and in some basal iguanodontian and ankylopollexian taxa (Norman, 2004). This foramen passes though the base of the jugal wing and opens on the anteromedial surface of the quadrate (Fig. 8D). The dorsal head of the quadrate is posteriorly recurved (Figs. 8C and 8D), unlike in the neornithischian Agilisaurus (Peng, 1992).

Two processes are present on the quadrate, the anteriorly directed jugal wing (Fig. 8D: jw) and the anteromedially directed pterygoid wing (Fig. 8C: pw). The jugal wing is a mediolaterally thin sheet that arises from the anterolateral margin of the quadrate shaft, is moderately developed, and extends ventrally nearly to the distal condyles (Fig. 8D), contrasting with the shortened, more dorsally situated jugal wing in some basal iguanodontians (e.g., Gasparinisaura, Zalmoxes: Coria & Salgado, 1996; Weishampel et al., 2003). A shallow fossa is present on the lateral surface of the quadrate shaft, just posterodorsal to the jugal wing. The pterygoid wing emerges from the anteromedial margin and is a large, anteromedially oriented sheet that arises dorsally below the head of the quadrate and ends well dorsal to the distal condyles (Fig. 8C). A fossa is present on the posterior side at the base of the pterygoid wing (Fig. 8C: pwf), which is also seen in the neornithischians Jeholosaurus, Parksosaurus, Orodromeus, and Zephyrosaurus (Galton, 1973; Sues, 1980; Scheetz, 1999; Barrett & Han, 2009; C Boyd, pers. obs., 2011) and the basal iguanodontian Dysalotosaurus (Norman, 2004). The anteroventral margin of the pterygoid wing is grooved where it inserted into a groove on the ventrolateral surface of the quadrate process of the pterygoid (Fig. 8E: pwg).

Description of the dorsal and ventral ends of the quadrate are based on the disarticulated right quadrate from TLAM.BA.2014.027.0001. The dorsal head of the quadrate is triangular in dorsal view. There is a slight ridge extending ventral from the posterolateral corner of the quadrate head that tapers ventrally and is lost less than a quarter of the way down the shaft. The pterygoid wing extends all the way to the dorsal end of the quadrate, forming the anteromedial corner. A thin ride extends up from the jugal wing, forming the anterolateral corner. The medial surface of the head is broadly rounded. A slight sulcus is present anteriorly below the dorsal head between the jugal and pterygoid wings.

The ventral end of the quadrate is slightly reniform (concave posteriorly) in ventral view. The distal condyles are not well-separated from each other. The lateral condyle extends further ventrally and is rounded ventrally. A sharp ridge extends from the jugal wing onto the anterolateral corner of the lateral condyle. The medial condyle is flattened ventrally and projects medially from the shaft of the quadrate. The pterygoid wing does not extend all the way to the medial distal condyle.

Pterygoid

The pterygoid consists of three processes oriented roughly orthogonal to each other: the quadrate process (Fig. 9E: qap), the mandibular process (Fig. 9E: mpp), and the palatine process (Fig. 9E: ppp). The quadrate process is a broad, dorsoventrally expanded, mediolaterally thin sheet that projects posterolaterally from the body of the pterygoid (Figs. 9E–9G). The posteromedial surface of the quadrate process is dorsoventrally concave (Fig. 9F). In the anterior corner of the quadrate process where it joins with the other processes a posteromedially facing cup is present that received the basipterygoid process of the basisphenoid (Fig. 9G: bpa). The anterolaterally facing surface of the quadrate process is dorsoventrally convex and contacted the pterygoid wing of the quadrate (Fig. 9E). The ventral margin of the quadrate process is mediolaterally thickened, creating an expanded ridge just ventral to an associated shallow groove that received the ventral edge of the pterygoid wing of the quadrate (Fig. 9E: lpr). A groove extends from the anterior edge of the lateral pterygoid ridge on the quadrate process ventrally onto the mandibular process (Fig. 9E: pg), as seen in Zephyrosaurus and an unnamed taxon from the Kaiparowits Formation of Utah (Sues, 1980; Boyd, 2012; Gates et al., 2013). This groove is absent in Thescelosaurus assiniboiensis (Brown, Boyd & Russell, 2011). The mandibular process projects ventrolaterally from the base of the quadrate process. Its surface is anteroposteriorly concave, with nearly the entire dorsal surface forming the articulation surface for the ectopterygoid (Fig. 9C). A thickened ridge is present along the anterior, lateral, and posterior margins of the mandibular process (Fig. 9E). The palatine process extends anteriorly from the contact between the mandibular and quadrate processes. At its base, a narrow shelf projects off the ventrolaterally from the ventral margin, forming the articulation surface for the palatine (Fig. 9E: apa). The remainder of the palatine process consists of a dorsoventrally expanded, mediolaterally thin process that extends anterodorsally from the body of the pterygoid. The distal end of the palatine process curves ventrally, eventually contacting the posterior process of the vomer. The palatal processes of the pterygoids were separated along the midline by a narrow interpterygoid vacuity (sensu Sereno, 1991). The dorsal margin of the palatal process continues posteriorly as a thin ridge that curves medial to the quadrate process, creating a dorsally flattened, medially projecting tab that contacting its antimere (Fig. 9G). The pterygoid is excluded from bordering the postpalatine fenestra by the ectopterygoid and palatine (Fig. 9C), as occurs in the neornithischians Haya and basal ceratopsians (Makovicky et al., 2011), but not in the neornithischians Changchunsaurus and Hypsilophodon (Galton, 1974a; Jin et al., 2010).

Palatine

The palatines are preserved slightly displaced from their natural positions, but are undamaged (Figs. 9A–9D). The palatine is robust laterally where it contacts the medial surface of the maxilla just dorsal to the posterior end of the tooth row. The contact surface for the maxilla is deeply dorsoventrally concave and relatively anteroposteriorly straight (Fig. 10A: am). Dorsal to the maxillary contact surface, a robust anterodorsally oriented projection is present that extends along the medial surface of the maxilla. The posterodorsal margin of this anterodorsal projection (Fig. 10A: adp) forms a broad contact surface with the anterior margin of the lateral process of the ectopterygoid (Fig. 9C). The dorsal tip of this projection inserts into the anteroposteriorly oriented groove in the medial surface of the anterior process of the jugal, as in Hypsilophodon and Lesothosaurus (Galton, 1974a; Sereno, 1991), preventing the palatine from contacting the lacrimal. In Jeholosaurus, the lacrimal is more ventrally positioned, inserting anterior to the jugal, allowing for a broad contact between the palatine and the lacrimal (Barrett & Han, 2009). Similarly, in Hypsilophodon the anterior process of the jugal is both dorsoventrally and anteroposteriorly shorter and does not extend as far anteriorly as in T. neglectus, facilitating more contact between the lacrimal and the palatine (Galton, 1974a).

A broad sheet extends dorsomedially from the thickened lateral surface, forming the ventromedial wall of the orbit (Figs. 2 and 10A). The dorsolateral surface of this sheet is mediolaterally convex and anteroposteriorly concave (Fig. 10A). A few low, mediolaterally oriented ridges are present on the dorsolateral surface that extend to the dorsal margin. The anterior margin of the palatine is slightly dorsoventrally thickened and has a ‘W-shaped’ outline in dorsal view, owing to the presence of a triangular anterior projection near the midpoint of the otherwise mediolaterally concave anterior margin (Figs. 10A and 10C). The anteromedial corner of the palatine consists of a thickened, ‘tab-shaped’ projection. The medial margin of the palatine is dorsoventrally thickened (more pronounced anteriorly) and relatively straight (Fig. 10A).

A deep sulcus is present in the posterolateral corner of the palatine (Figs. 10A and 10C: ppf). This sulcus formed the anterior, and part of the medial margin of the postpalatine fenestra (sensu Sereno, 1991: = suborbital fenestra of Makovicky et al., 2011). Medial to this sulcus the posterior margin is slightly mediolaterally concave and angles anteromedially. The posterodorsal corner of the palatine is rounded. The ventromedial surface of the palatine is mediolaterally and anteroposteriorly concave (Fig. 10C). The articulation surface for the palatal process of the pterygoid consists of a flattened facet on the ventromedial surface positioned just medial to the sulcus for the postpalatine fenestra (Figs. 10B and 10C: apt). The palatines may have contacted each other along at least part of their medial margins, as in Orodromeus (Scheetz, 1999). It does not appear that the palatines extended far enough anteriorly to contact the posterolateral processes of the vomer (Figs. 9A–9D), as it does in Hypsilophodon and Lesothosaurus (Galton, 1974a; Sereno, 1991). The palatines of NCSM 15728 match the morphology of the highly fragmentary palatines preserved in the holotype of Thescelosaurus assiniboiensis (Brown, Boyd & Russell, 2011: Fig. 9).

Ectopterygoid

The medial portion of the ectopterygoid consists of an expanded plate with a ventromedially facing articulation surface that contacts nearly the entire dorsolateral surface of the mandibular process of the pterygoid (Fig. 10F). The ectopterygoid did not contact the palatal process of the pterygoid, as it does Changchunsaurus (Jin et al., 2010), though its anteromedial corner does just touch the posteromedial corner of the palatine at the base of the palatal process. The anteromedial portion of the ectopterygoid formed the medial margin of the postpalatine fenestra. In ventromedial view the articulation surface for the pterygoid is roughly triangular in shape, with its apex pointed anteromedially.

A ‘rod-shaped’ lateral process extends from the dorsolateral surface of the medial plate and angles anterodorsally (Figs. 10D and 10F). This process is bowed along its length, being convex dorsally and concave ventrally (Fig. 10E). A small fenestra is present between the maxilla and the ventral surface of this lateral process. The lateral process of the ectopterygoid becomes anteroposteriorly wider and dorsoventrally thinner dorsally, as it curves around the posteromedial corner of the maxilla. Near the dorsal end of the lateral process the anteroventral surface overlaps the posterodorsal corner of the maxilla. The distal end of the lateral process bears an anteroposteriorly oriented ridge that inserted into a dorsoventrally narrow groove on the medial surface of the anterior process of the jugal (Figs. 10D and 10E), just dorsal to the posterior end of the maxilla and posterior to the palatine, which also inserts into the groove.

The dorsal half of the anterior margin of the lateral process formed a broad, concave articulation surface for the posterior margin of the anterodorsal projection of the palatine (Fig. 10F), though postmortem displacement of the palatines has removed these elements from contact in NSCM 15728 (Fig. 9C). Contact between the ectopterygoid and the palatine is also present in Haya and Lesothosaurus (Sereno, 1991; Makovicky et al., 2011), but is apparently absent in Changchunsaurus and Hypsilophodon (Galton, 1974a; Jin et al., 2010). The contact between the lateral process of the ectopterygoid and the palatine terminated medially at the anterolateral corner of the postpalatine fenestra, and together these two bones formed the entire border of this fenestra.

Vomer

The vomer is a midline element that contacted the premaxillae and maxillae anteriorly and the pterygoid posteriorly. The anterior end of the vomer is triangular in dorsal view (Fig. 9I), is dorsoventrally thin, and overlapped the maxillae along their ventral surfaces. The anterior tip inserted into a short socket near the ventral margin of the premaxillae. Posterior to the triangular anterior end a mediolaterally narrow neck is present that angles posterodorsally, separating the maxillae along the midline. Shortly after passing between the maxillae the vomer turns posteriorly. A narrow groove indents the dorsal surface of the vomer, and mediolaterally thin, dorsolaterally oriented processes arise from the dorsolateral margins of this groove (Fig. 9I). Posteriorly, the dorsal groove deepens and the dorsolateral processes become more dorsoventrally elongate.

A mediolaterally thin process extends from the ventral margin, becoming more elongate posteriorly, which makes the posterior portion of the vomer ‘Y-shaped’ in transverse section (Fig. 9H). A portion of this ventral process is damaged and was lost (Fig. 9H: d). Near their posterior end the dorsal groove extends all the way through the ventral process, dividing the posterior end into two lateral wings (Fig. 9I: ppv). The dorsal margins of these lateral wings become mediolaterally thicker as they extend posteriorly, and eventually the wings overlap the lateral surfaces of the palatal processes of the pterygoids (sensu Sereno, 1991) as occurs in all ornithischian taxa (Sereno, 1991). The palatal process of the pterygoid is not reconstructed in contact with the posterior end of the vomer in Hypsilophodon (Galton, 1974a), but in all specimens of Hypsilophodon the palatal process is largely missing and this may not be accurate. The posterior extent of the vomer is roughly equal with the anterior margin of the orbit; thus, the vomer could not have contacted the palatines (Figs. 9A–9D).

Braincase

The braincase of NSCM 15728 is slightly transversely crushed (Fig. 4), and demonstrates a lack of fusion between most of the individual bones (Fig. 11), with a few exceptions. Each opisthotic is fused indistinguishably with its exoccipital (Figs. 11A, 11B, 13A–13D), as is the general case in basal ornithischians (Norman et al., 2004), and the parasphenoid and the basisphenoid are indistinguishably fused (Figs. 11A, 11B, 11D, 13G–13I), which occurs in nearly all dinosaurs (Currie, 1997). The left opisthotic/exoccipital is slightly inset medially from its normal position (Fig. 11E), while the right opisthotic/exoccipital is displaced anteriorly and slightly medially (Fig. 4). The prootics, laterosphenoids, and the supraoccipital are all slightly displaced anteriorly, so that small gaps are present between each of these bones and most of their adjacent bones (Figs. 11A and 11B). There is no evidence of an ossified orbitosphenoid or presphenoid in NCSM 15728. The presence of an orbitosphenoid was noted in Parksosaurus (Galton, 1973: Figs. 2 and 3); however, its placement and morphology suggest that this is actually a slightly damaged palatine (C Boyd, pers. obs., 2011). This observation is supported by the fact that this bone was reconstructed in the same position the palatine occupies in NCSM 15728 (Galton, 1973: Fig. 5). Thus, the lack of an ossified orbitosphenoid in NCSM 15728 is not unexpected. In TLAM.BA.2014.027.0001 there are some fragmentary pieces of bone positioned anterior to the dorsal half of the right laterosphenoid that may represent part of an ossified orbitosphenoid, but exact identification of those fragments is difficult given their poor preservation.

Basioccipital

The left posterodorsal corner of the basioccipital is detached from the rest of the basioccipital and preserved on the block containing the anterior cervical vertebrae from NCSM15728 (Figs. 12G and 12I). The anteroposterior length of the basioccipital is greater than the length of the basisphenoid, not including the fused parasphenoid (Fig. 11D), as in the neornithischians Jeholosaurus and Thescelosaurus assiniboiensis (Barrett & Han, 2009; Brown, Boyd & Russell, 2011) and the iguanodontians Camptosaurus, Dryosaurus, and Tenontosaurus tilletti (Norman, 2004). The posterior surface of the basioccipital forms the majority of the occipital condyle, along with contributions from the posteromedial portion of the fused opisthotic/exoccipital (Fig. 11E). The posterodorsal surface of the basioccipital is indented by the ventral margin of the foramen magnum, which occupies between twenty and thirty percent of the posterodorsal surface of the basioccipital (Fig. 11E), as in the neornithischians Orodromeus, Oryctodromeus, Othnielosaurus, and Zephyrosaurus (Sues, 1980; Scheetz, 1999; C Boyd, pers. obs., 2011) and the basal iguanodontian Zalmoxes (Weishampel et al., 2003). In Thescelosaurus assiniboiensis the ventral margin of the foramen magnum occupies more than one third of the posterodorsal surface of the basioccipital (Brown, Boyd & Russell, 2011).

The lateral and ventral sides of the basioccipital are concave, giving the basioccipital an ‘hour-glass’ shape in ventral and lateral views (Figs. 12G and 12H). A ventrally extending keel is present along the midline of the ventral surface of the basioccipital, extending from the anterior contact with the basisphenoid to about one third of the way toward to posterior end (Fig. 12G: bk). A similar keel is present in all non-iguanodontian neornithischian taxa except Othnielosaurus (Scheetz, 1999). Immediately posterior to this keel a small foramen is present in NCSM 15728 (Fig. 12G: bf), which is not known in any other neornithischian taxon (Norman et al., 2004). This foramen penetrates dorsally into the basioccipital, but does not appear to penetrate the floor of the braincase based on examination of the CT data. This foramen is not present in TLAM.BA.2014.027.0001, and may represent individual variation within this species. Lateral to the ventral keel along the anterior margin of the basioccipital, two small knobs form the posterior portions of the basal tubera (Figs. 12G and 12H). These basioccipital contributions to the basal tubera extend ventrally to the same level as their counterparts from the basisphenoid (Figs. 11A and 11B).

The anterior margin of the basioccipital forms an anteriorly pointing ‘V-shape’ in ventral view, inserting into the body of the basisphenoid (Figs. 11D and 12G). This results in the articulation surface for the basisphenoid extending onto the anterolateral surfaces of the basioccipital (Figs. 11A, 2, 11B, 11D and 12H), creating a tightly interlocking contact between these two elements. A similar morphology is seen in the neornithischians Changchunsaurus and Haya (Jin et al., 2010; Makovicky et al., 2011) and the basal iguanodontian Anabisetia (Coria & Calvo, 2002). Alternatively, the anterior margin of the basioccipital is ‘W-shaped’ (indented posteriorly along the midline) in Zephyrosaurs (Sues, 1980), and is relatively flat in Thescelosaurus assiniboiensis (Brown, Boyd & Russell, 2011). In Hypsilophodon and Parksosaurus the basioccipital and basisphenoid are indistinguishably fused, obscuring the shape of their mutual contact (Galton, 1973; Galton, 1974a). The lateral portions of the dorsal surface (Fig. 12I) form rugose articulation surfaces for the fused opisthotic/exoccipital (posteriorly) and the prootic (anteriorly). Between these articulation surfaces the medial third of the dorsal surface of the basioccipital is slightly depressed and roughly ‘hour-glass’ shaped, forming the posterior portion of the floor of the braincase. The anterior portion of this depressed surface is dorsally arched along the midline (Fig. 12: afb), as in all non-iguanodontian neornithischian taxa except Othnielosaurus (Scheetz, 1999).

Basisphenoid/Parasphenoid

The anteroposterior length of the basisphenoid, not including the parasphenoid, is shorter than the length of the basioccipital (Fig. 11D). The posterodorsal surface of the basisphenoid formed the anteroventral floor of the braincase (Fig. 13J). The posteroventral margin of the basisphenoid contribution to the floor of the braincase extended posteriorly and slightly overlapped the dorsal surface of the basioccipital (Fig. 13I). The median ridge on the anterodorsal surface of the basioccipital extends onto the basisphenoid. On either side of this median ridge a shallow groove is present that deepens posteriorly, eventually connecting to a set of foramina that penetrate the floor of the braincase and pass into the posterodorsal surface of the sella turcica (Fig. 13J). The abducens nerve (CN VI) passed through these foramina. Lateral and slightly ventral to the basisphenoid contribution to the floor of the braincase, the laterally projected preotic pendants are present (Figs. 13G–13K: prp). The dorsal surfaces of the preotic pendants face posterodorsally and formed part of the articulation surface for the prootic (Figs. 11A and 11B). Their lateral margins are slightly dorsoventrally concave. A sharp ridge marks the ventrolateral margin of these processes, and the surface ventral to this ridge is dorsoventrally concave (Fig. 13K). This concavity deepens posteriorly, forming a mediolaterally narrow fossa posteroventral to each preotic pendant.

The posterior surface of the basisphenoid forms a ‘V-shaped’ (in ventral view) contact with the anterior surface of the basioccipital (Fig. 13H), with the posterolateral margins of the basisphenoid extending onto the posterolateral surfaces of the basioccipital. The basisphenoid contribution to the basal tubera was level with the corresponding contribution from the basioccipital (Figs. 11A and 11B). The lateral surfaces of the basisphenoid were concave anteroposteriorly and convex dorsoventrally posterior to the basipterygoid processes (Fig. 13H). The ventral surface is slightly anteroposteriorly concave and mediolaterally convex anterior to the basipterygoid processes. Between the basipterygoid processes the ventral surface is mediolaterally and anteroposteriorly concave. The basipterygoid processes are stout and project anterolaterally from the ventrolateral corners of the basisphenoid (Figs. 13G–13I and 13K). These processes are oriented approximately sixty degrees from each other (Figs. 13H and 13K). The posterior margins of the basipterygoid processes are mediolaterally convex, while the anterior edges consist of sharply rounded ridges. The anteroventral surfaces of the basipterygoid processes are mediolaterally and dorsoventrally flattened, forming contact surfaces for the pterygoids (Figs. 13H, 13I and 13K).

The cutriform process arises just anterior to the bases of the basipterygoid processes and projects anteriorly along the midline (Figs. 13G, 13H and 13I), with the anterior tip extending between the palatines. Mediolateral compression of the specimen caused the palatines to compress and damage the anterior end of the cutriform process (Figs. 13G and 13H). The ventral surface of the cutriform process bears a distinct ventral ridge that deepens posteriorly until just anterior to the basipterygoid processes, at which point the ventral ridge decreases in dorsoventral height until it ends approximately level with the anterior margin of the basipterygoid processes. This gives the ventral margin of the cutriform process a triangular shape in lateral view (Fig. 13I). The dorsal surface of the cutriform process is mediolaterally concave, with the dorsolateral projections on either side of the dorsal concavity becoming dorsoventrally taller and more vertically oriented posteriorly (Fig. 13G). The dorsolateral margins contact each other just anterior to the sella turcica, creating a short foramen (Fig. 13G). This foramen passes into the anteroventral surface of the sella turcica.

The sella turcica is enclosed within the anterodorsal portion of the basisphenoid (Fig. 13G). The foramina for the internal carotid arteries pass through the posteroventral corners of the sella turcica, exiting in the fossa ventral to the preotic pendants. Many authors have speculated that the foramina for the internal carotid arteries were present ventral to the preotic pendants (e.g., Galton, 1974a; Galton, 1989; Sues, 1980; Sereno, 1991; Butler, Smith & Norman, 2007), but their presence was never previously observed either via visual inspection or examination of CT data (Butler, 2010). The CT data collected from NCSM 15728 confirms the presence of these foramina ventral to the preotic pendants in at least this taxon. The fused basisphenoid/parasphenoid of T. neglectus differs substantially from that of the neornithischian Zephyrosaurus. In the latter taxon the basipterygoid processes projected ventrally, but not anteriorly (Sues, 1980). The ventral surface of the basisphenoid bears an elongate groove extending from the posterior contact with the basioccipital to the base of the cutriform process anteriorly (Sues, 1980). Additionally, the posterolateral surface of the basisphenoid bears a prominent depression (Sues, 1980), which is lacking in T. neglectus.

Opisthotic/Exoccipital

The ventral margin of the fused opisthotic/exoccipital formed an extensive contact with the dorsolateral surface of the basioccipital (Figs. 13C–13D). The posteroventral portion of the fused opisthotic/exoccipital forms the dorsolateral corner of the occipital condyle (Fig. 13A: oc), unlike in ankylopollexians where the exoccipital is excluded from the occipital condyle (Norman, 2004). The ventral margin of this posterior process is mediolaterally wider than the dorsal margin, and the dorsal edge is mediolaterally convex (Fig. 13A). The medial surface of the fused opisthotic/exoccipital is broadly dorsoventrally concave, forming the posterolateral wall of the braincase (Fig. 13A). A small, shallow fossa is present on the medial surface for the remnant of the vena cerebralis posterior (Galton, 1989). The posteromedial margin of the fused opisthotic/exoccipital forms the majority of the foramen magnum. On the posterior surface of the dorsomedial corner of the fused opisthotic/exoccipital a posteriorly projecting boss is present (Figs. 13A and 13D: pbro) that served as the articulation surface for the proatlas (Sereno, 1991). The anterodorsal surface of the fused opisthotic/exoccipital forms a complex articulation surface with the supraoccipital that is pierced by the foramen for the posterior semicircular canal. The anterior margin formed an extensive contact with the prootic, and the foramen for the lateral semicircular canal is present on the dorsal portion of this contact. Medial to the foramen for the lateral semicircular canal a fossa is present that formed the posterior portion of the vestibule (Fig. 13B).

The paroccipital process arises from the dorsolateral body of the fused opisthotic/exoccipital and extends dorsolaterally (Fig. 13A). The paroccipital process is anteroposteriorly thinner than anteroventrally tall. The distal end of the paroccipital process expands ventrally, giving it a ‘pendent-shape’ in posterior view (Fig. 13A), unlike in the basal ornithischian Lesothosaurus (Sereno, 1991), basal thyreophorans (Norman, Witmer & Weishampel, 2004b), and the neornithischians Agilisaurus, Gasparinisaura, and Hypsilophodon (Galton, 1974a; Peng, 1992; Coria & Salgado, 1996) where the distal end is at most slightly widened. There is no enclosed posttemporal foramen in the paroccipital process; instead, the dorsal margin of the paroccipital process is notched by a ‘Y-shaped’ groove that is open dorsally through which passed the vena capitis dorsalis (Figs. 13A and 13B), as in Thescelosaurus assiniboiensis (Brown, Boyd & Russell, 2011). This groove begins on the posteromedial surface of the paroccipital process and extends anteromedially over the dorsal margin of the neck of the paroccipital process. At the apex of the dorsal margin the paroccipital process this groove bifurcates, with one branch passing anteriorly through a deep, nearly enclosed groove and the other branch angling dorsomedially to the contact with the supraoccipital. This latter groove continued onto the dorsal margin of the supraoccipital, while the former penetrates the medial process of the squamosal. The anterior surface of the paroccipital process broadly contacted the posterior surface of squamosal.

The crista tuberalis is a prominent ridge that arises along the ventral margin of the paroccipital process and extends anteroventrally to the contact with the basioccipital (Figs. 13C and 13D). Anterior to the crista tuberalis portions of three foramina/fenestra are present. The anterior-most of these is the posterior margin of the fenestra ovalis (Figs. 13C and 13D), into which the stapes inserts. Posterior to the fenestra ovalis an anteroventrally projecting process, the crista interfenestralis, separates the fenestra ovalis from the foramen metoticum (Figs. 13C and 13D). The foramen metoticum facilitates the passage of the glossopharyngeal nerve (CN IX), the accessory nerve (CN XI), and the vena jugularis interna from the braincase (Galton, 1989). Dorsal to these foramina the anteroventral surface of the paroccipital process is very shallowly concave (Fig. 13C) and lacks the dorsoventrally deep and anteroposteriorly narrow lateral opisthotic fossa seen in Orodromeus, Zephyrosaurus, and an unnamed taxon from the Kaiparowits Formation of Utah (Sues, 1980; Scheetz, 1999; Boyd, 2012; Gates et al., 2013). A foramen present on the anterior surface of the crista tuberalis (Fig. 13B) passes posteriorly through this ridge, emerging on its posterior surface (Fig. 13C). The vagus nerve (CN X) exits the braincase via the foramen metoticum and then passes posteriorly through this foramen. Posterior to the crista tuberalis, the ventrolateral surface of the fused opisthotic/exoccipital is pierced by two closely spaced foramina (Fig. 13C). The more dorsally positioned foramen penetrates medially to the lateral wall of the braincase and housed the posterior ramus of the hypoglossal nerve (CN XII). The ventral-most foramen also penetrated directly medially to the lateral wall of the braincase and housed the anterior ramus of the hypoglossal nerve (Fig. 13D). In the neornithischian Jeholosaurus only a single foramen is present on the lateral surface for the passage of the hypoglossal nerve (Barrett & Han, 2009).

Prootic

The prootic formed the lateral wall of the braincase and was bordered posteriorly by the opisthotic, posteroventrally by the basioccipital, anteroventrally by the basisphenoid, anteriorly by the laterosphenoid, and dorsally by the supraoccipital (Figs. 11A and 11B). The dorsal margin is slightly anteroposteriorly concave, with the posterior end rising dorsally to overlap the posterodorsal surface of the fused opisthotic/exoccipital (Fig. 11A). The dorsal articulation surface for the supraoccipital is roughly triangular in shape, being mediolaterally broad posteriorly and narrowing anteriorly, and is pierced by the foramen for the anterior semicircular canal. The anterodorsal corner or the prootic bears a dorsoventrally elongate projection that inserted into the posterior end of the laterosphenoid (Fig. 13E). The anteroventral corner of the prootic is broadly rounded to contact the basisphenoid.

The lateral surface of the prootic is pierced by two foramina (Fig. 13E). The trigeminal, or prootic, foramen for CN V (the trigeminal nerve) is located near the anterior end of the prootic and is entirely enclosed within the prootic (Fig. 13E), unlike in Thescelosaurus assiniboiensis (Brown, Boyd & Russell, 2011). A narrow groove extends from the anterodorsal corner of this foramen and extends anterodorsally onto the posteroventral margin of the laterosphenoid (Fig. 13E). The ramus ophthalamicus (CN V 1) of the trigeminal nerve likely passed anteriorly through this groove before passing through a similar groove in the laterosphenoid. A narrow ledge projects over the dorsal margin of the trigeminal foramen, beginning at the anterior margin of the prootic and extending posteriorly about half the length of the prootic. The facialis foramen (sensu Galton, 1989) for CN VII is a relatively small foramen positioned posteroventral to the trigeminal foramen (Fig. 13E). A flat sheet of bone extends posterolaterally between these foramina, laterally overhanging the facialis foramen. This sheet of bone runs anteroventrally and is confluent with the posterolateral margin of the basipterygoid process of the basisphenoid. A depression extends ventral to the facialis foramen medial to this sheet, through which passed the ramus palatines (CNV IIp: Galton, 1989). The posteroventral corner of the prootic extends ventral to the fenestra ovalis to contact the dorsal margin of the basioccipital and the anteroventral corner of the fused opisthotic/exoccipital. The crista prootica (Fig. 13E: cpr) forms a sharp edge overhanging the anterodorsal margin of the foramen ovale (Fig. 13E: fo) and, more ventrally, the lateral surface of the lagenar recess. The fenestra ovalis, into which the stapes inserts, notches the posterior margin of the prootic at approximately mid-height, with the fused opisthotic/exoccipital forming the posterior margin of this fenestra (Fig. 11A). Just ventral to the fenestra ovalis the posterior margin is indented to form the ventral margin of the foramen metoticum (Fig. 13E). Dorsal to the fenestra ovalis, the posterodorsal contact surface for the fused opisthotic/exoccipital is penetrated by the foramen for the lateral semicircular canal.

On the dorsomedial surface of the prootic, near the suture for the supraoccipital, the shallow, anteriorly facing fossa subarcuata is present (Fig. 13F: fs). Near the anterior margin of the medial surface the foramen for the trigeminal nerve is present. Extending from the anteromedial margin of the trigeminal foramen is a deeply recessed groove that runs anteriorly to the contact with the ventral edge of the laterosphenoid (Fig. 13F: vcms). It is likely that at least the vena cerebralis media secunda, if not the entire vena cerebralis media, occupied this medial groove and exited the foramen at the posteroventral margin of the laterosphenoid. The ramus ophthalamicus (CN V 1) of the trigeminal nerve likely did not pass through this groove; rather, it occupied the groove on the lateral surface of the prootic (Fig. 13E). Posterior to the trigeminal foramen, a fossa is present on the medial surface that contains three foramina (Fig. 13F), as in Dysalotosaurus (Galton, 1989).The facialis foramen for CN VII passes laterally out of the anteroventral corner of this fossa. The foramen for the anterior ramus of the acoustic nerve (Fig. 13F: CN V IIIa) is positioned in the anterodorsal portion of this fossa and travels dorsolaterally into the anterior utricular recess within the prootic. The foramen for the posterior ramus of the acoustic nerve (CN V IIIp) is positioned posterodorsally in the fossa and extends posterodorsally into the lagenar recess. This differs from the morphology seen in the neornithischian Hypsilophodon and the basal iguanodontian Dryosaurus where a fossa does not connect the former two foramina with the foramen for the posterior ramus of the acoustic nerve (Galton, 1989).

Stapes

The stapes is a ‘rod-shaped’ bone that extends from the fenestra ovalis (formed by the fused opisthotic/exoccipital and the prootic) to the anterolateral surface of the paroccipital process, medial to the dorsal end of the quadrate (Fig. 11A), the presumed location of the otic notch (You & Dodson, 2004). The proximal end is broadened dorsoventrally where it enters the fenestra ovalis, with a thin ridge present on the dorsal surface. The full morphology of the proximal end cannot be determined because it is closely appressed to the braincase and difficult to fully distinguish in the CT scans and is covered by matrix, making it impossible to fully describe via visual examination. After exiting the fenestra ovalis, the stapes angles posteroventrally, and slightly dorsally as well. The mid-shaft portion is rounded in transverse section. Near the distal end the stapes narrows mediolaterally while expanding dorsoventrally, giving the distal end a triangular shape in lateral view.

The morphology of the stapes in NCSM 15728 closely matches that figured (though not described) for the basal ornithischian Lesothosaurus by Sereno (1991). Among neornithischians, a stapes is only identified for a single specimen of Jeholosaurus (IVPP V15716: Barrett & Han, 2009). However, the position (near the distal condyles of the quadrate pointing posteriorly) and morphology of the presumed stapes in the referred specimen of Jeholosaurus (Barrett & Han, 2009: Fig. 6) matches that of the element identified as a ceratobranchial in the holotype of Changchunsaurus (Jin et al., 2010: Fig. 1B). In the same specimen of Jeholosaurus, a slender, ‘rod-shaped’ element exposed displaced in the infratemporal fenestra was identified as a possible epipterygoid (Barrett & Han, 2009: Fig. 6). Among ornithischian dinosaurs, ossified epipterygoids are only known in a few ankylosaurian and pachycephalosaurian dinosaurs (Maryańska, Chapman & Weishampel, 2004; Vickaryous, Maryańska & Weishampel, 2004). Thus, it is possible that the element identified as an epipterygoid in the referred specimen of Jeholosaurus (IVPP V15716) is either a slightly displaced stapes or a hyoid element, especially given that the specimen is extensively transversely crushed. A slender element in the holotype of Jeholosaurus positioned dorsolateral to the basisphenoid and exposed in ventral view was also identified as a possible epipterygoid (Barrett & Han, 2009). However, this bone is triangular to ‘T-shaped’ in cross section, owing to the presence of a ventrally projecting ridge and concave lateral and medial surfaces, and appears to be the tip of a larger element that is obscured by the basisphenoid (C Boyd, pers. obs., 2011). This morphology does not match the morphology of any epipterygoid previously reported for an ornithischian dinosaur (Maryańska, Chapman & Weishampel, 2004; Vickaryous, Maryańska & Weishampel, 2004) or that of the stapes, and it seems likely this exposed end of bone is part of one of the bones of the palate. A stapes is preserved in original position in another referred (but undescribed) specimen of Jeholosaurus (PKUP V 1601), and its morphology generally conforms with that here reported in NCSM 15728 (C Boyd, pers. obs., 2011).

Laterosphenoid

The laterosphenoid contacted the prootic along an obliquely inclined surface along it posteroventral margin (Fig. 11A). A dorsoventrally elongate, mediolaterally thin dorsomedial process of the prootic inserted into the posterior end of the laterosphenoid. There is no evidence of the posteroventrally projected prootic boss along the contact surface that inserted into the prootic that is seen in an unnamed taxon from the Kaiparowits Formation of Utah (Boyd, 2012; Gates et al., 2013). The trigeminal, or prootic, foramen for CN V does not notch the posterior end of the laterosphenoid, unlike in the heterodontosaurid Heterodontosaurus (Norman et al., 2004), the basal ornithischian Lesothosaurus (Sereno, 1991), the basal ornithischians Gasparinisaura, Hypsilophodon, Jeholosaurus, Parksosaurus, Thescelosaurus assiniboiensis, and Zephyrosaurus (Galton, 1973; Galton, 1974a; Sues, 1980; Coria & Salgado, 1996; Brown, Boyd & Russell, 2011; C Boyd, pers. obs., 2011), the basal iguanodontians Tenontosaurus and Zalmoxes (Norman, 2004; Weishampel et al., 2003), and the ankylopollexians Camptosaurus and Iguanodon (Norman, 2004).

On the posteroventral corner of the anterior surface of the laterosphenoid, along the contact with the prootic, a deep groove is present through which a portion of the trigeminal nerve (CN V), the ramus ophthalamicus, (CN V 1: Galton, 1989) passed through after exiting the prootic foramen and traveling through an anterodorsally oriented groove on the lateral surface of the prootic (Figs. 12A–12D: vlg). A similar groove may be present in the basal ornithischian Lesothosaurus (Sereno, 1991) and is seen in the neornithischians Orodromeus and Zephyrosaurus (Sues, 1980; Scheetz, 1999). Just ventromedial to this groove the dorsal edge of a foramen notches the ventral margin of the laterosphenoid, along the contact with the prootic (Fig. 12D: vcms). The vena cerebralis media passes through a channel between the prootic and the laterosphenoid anterodorsal to the trigeminal foramen in the neornithischian Zephyrosaurus and the basal iguanodontian Dryosaurus (Galton, 1989), with the vena cerebralis media secunda then passing anteriorly through a groove in the posteroventral margin of the laterosphenoid. Thus, it is likely that at least the vena cerebralis media secunda passed through this foramen in T. neglectus, if not the entire vena cerebralis media.

The lateral surface of the laterosphenoid is dorsoventrally convex and anteroposteriorly concave and lacks the foramen in the posteroventral corner seen in Lesothosaurus that is hypothesized as the foramen for the oculomotor nerve (CN III: Sereno, 1991). The dorsal margin articulates with the ventral margin of the parietal, and this contact consists of a sharp ridge anteriorly that becomes mediolaterally broader posteriorly and bears a series of low ridges and grooves that interlocks with corresponding ridges and grooves on the parietal. This ridge extends anterior to the medial margin of the head of the laterosphenoid. The anterodorsal head of the laterosphenoid turns laterally (Fig. 12D), forming a broad contact surface with the frontal and the postorbital, as in all neornithischians except Orodromeus (Scheetz, 1999) and in the basal iguanodontians Tenontosaurus dossi and Zalmoxes (Weishampel et al., 2003; Norman, 2004). The anterior margin of the head is concave (Fig. 12C), while the posteromedial and lateral margins are slightly convex (Figs. 12A and 12B).

The dorsomedial surface is broadly concave both dorsoventrally and anteroposteriorly (Fig. 12A). The medial margin consists of a rounded ridge that extends from the posteromedial corner of the laterosphenoid to the anterodorsal head. Midway along this ridge an expanded, anteromedially projected boss is present (Figs. 12A, 12C and 12D: ob). In Hypsilophodon, a similar boss, or step, on the anteromedial edge is hypothesized to demarcate the ventral extent of the unossified orbitosphenoid (Galton, 1974a), and in Dryosaurus a similar boss is present at the ventral margin of the contact between the laterosphenoid and the ossified orbitosphenoid (Galton, 1989). Just dorsal to this boss a semicircular depression is present, indicating the presence of a foramen that passed between the laterosphenoid and the orbitosphenoid (Figs. 12C and 12D). In Dryosaurus, the foramen for the trochlear nerve (CN IV) passes between the laterosphenoid and the orbitosphenoid dorsal to the medially projecting boss (Galton, 1989), making it likely that this foramen served the same function.

Supraoccipital

The posteroventral tip of the supraoccipital just barely contacts the dorsal margin of the foramen magnum, as seen in Thescelosaurus assiniboiensis (Brown, Boyd & Russell, 2011) and unlike all other basal ornithischians where the supraoccipital forms a substantial portion of the dorsal margin of the foramen magnum (Norman et al., 2004). A few small foramina pierce the dorsal surface of the posteromedial portion of the supraoccipital (Fig. 4); however, none of these foramina penetrate through to the ventral surface of the supraoccipital and do not represent the relatively large, medially situated supraoccipital foramen present in Thescelosaurus assiniboiensis (Brown, Boyd & Russell, 2011). Two narrow grooves extend along the dorsal surface of the supraoccipital, beginning at the posterolateral margins and extending anteromedially until they reach distinct foramina that pierce the dorsal surface of the supraoccipital. These grooves are continuations of the dorsally open grooves present on the fused opisthotic/exoccipital that contained vena capitis dorsalis, as seen in Thescelosaurus assiniboiensis (Brown, Boyd & Russell, 2011).

The dorsal process of the supraoccipital is triangular in lateral view (Fig. 12E), being dorsoventrally tall anteriorly and tapering posteriorly. This dorsal process inserted into the concave ventral surface of the parietal. The posterodorsal surface of the dorsal process is slightly mediolaterally convex, but lacks the distinct nuchal crest seen in the basal ornithischians Eocursor and Lesothosaurus (Sereno, 1991; Butler, 2010), the neornithischians Gasparinisaura and Orodromeus (Scheetz, 1999), the basal iguanodontian Tenontosaurus (Norman, 2004), and in some ankylopollexians (Norman, 2004). The lateral surfaces of this dorsal process are gently dorsoventrally concave (Fig. 12E).

The ventrolateral processes of the supraoccipital are relatively thin posteriorly, but thicken anteriorly. The posteroventral margin of the ventrolateral processes sutured to the dorsomedial surface of the fused opisthotic/exoccipital. Posterior to this contact, the ventral surface formed an elongate contact with the prootic. The anterior margins of the ventrolateral processes thin to form narrow, posterodorsally inclined ridges. The medial surface of the supraoccipital is deeply concave, forming the posterodorsal roof of the braincase. A separate or co-ossified epiotic forming an anterolaterally extending flange that was directed under the parietal wings, forming the posterodorsal part of the lateral wall of the braincase, which is seen in the basal ornithischians Eocursor and Lesothosaurus (Sereno, 1991; Butler, 2010), is absent in T. neglectus.

Dorsally directed fossae that formed the dorsal portions of the vestibules are present along the ventral margins of the ventrolateral processes of the supraoccipital, along the shared contacts between the supraoccipital, prootic, and fused opisthotic/exoccipital. On each side, the dorsomedial surface of the fossa for the vestibule is penetrated by the foramen for the crus communis that extends dorsally into the supraoccipital close to the medial surface. The foramen for the crus communis bifurcates dorsally into the foramina for the anterior and posterior semicircular canals. The foramen for the posterior semicircular canal does not extend as far dorsally into the supraoccipital as the foramen for the anterior semicircular canal. The foramen for the posterior semicircular canal exits the posteroventral surface of the ventrolateral process of the supraoccipital along the articulation surface with the fused opisthotic/exoccipital. The foramen for the anterior semicircular canal extends high into the supraoccipital, then arcs ventrolaterally, eventually exiting the ventral margin of the supraoccipital along the articulation surface for the prootic. The fossa subarcuata (sensu Galton, 1989), a depression in the ventromedial surface of the supraoccipital adjacent to the prootic articulation surface that housed the floccular lobe of the cerebellum, lies within the loop formed by the anterior semicircular canal as in the neornithischians Hypsilophodon and Zephyrosaurus and in basal iguanodontians (e.g., Dryosaurus, Tenontosaurus: Galton, 1989). The fossa subarcuata is greatly reduced relative to its development in the neornithischians Hypsilophodon and Orodromeus (Galton, 1974a; Scheetz, 1999), as also occurs in Thescelosaurus assiniboiensis (Brown, Boyd & Russell, 2011).

Mandible

The mandibles of NSCM15728 are well preserved and remain in close contact with the rest of the skull (Figs. 1 and 2). Unfortunately, this resulted in much of the dentary dentition being obscured by the overhanging maxillary dentition. The post-dentary elements on the left side of the skull are slightly displaced from their original positions, making it easier to define the boundaries of the individual elements. Therefore, the post-dentary elements from the left side were used for the figures.

Predentary

The anterior tip of the predentary is sharply pointed in ventral view (Fig. 14B) as in heterodontosaurids (e.g., Heterodontosaurus: Butler, Porro & Norman, 2008), neornithischians (e.g., Jeholosaurus, Changchunsaurus; Barrett & Han, 2009; Jin et al., 2010), and marginocephalians (e.g., Archaeoceratops, Liaoceratops, Yinlong: Xu et al., 2002; You & Dodson, 2003; Xu et al., 2006), and Hypsilophodon (Galton, 1974b), but unlike the rounded anterior margin seen in most basal iguanodontians (e.g., Tenontosaurus, Zalmoxes: Ostrom, 1970; Weishampel et al., 2003). The anteroventral surface is broadly rounded. A single, prominent foramen pierces the lateral surface of the main body of the predentary, though its placement varies slightly on each side (Figs. 14A and 14B: lfpd). Several smaller foramina are also present on the lateral surfaces, but their number and placement varies on each side. A broad, shallow groove extends posteroventrally along the lateral surface from near the anterior tip, passing through the lateral foramen, and ending near the middle of the embayment formed between the posterolateral and posteroventral processes that received the anterior tip of the dentary (Figs. 14A and 14B: lg). This groove is similar to the shallow groove seen in Hypsilophodon (Galton, 1974a), but not as prominent as the lateral groove seen in Changchunsaurus and Jeholosaurus (Barrett & Han, 2009; Jin et al., 2010). Prominent posterolateral processes extend posteriorly onto the dorsal surface of the anterior portion of the dentary (Figs. 14A and 14B: plpd) and are pierced at approximately mid-length by a foramen. The posterolateral processes are anteroposteriorly longer than dorsoventrally tall, and the posterior ends are bluntly pointed. The posterior end of the posterolateral processes are separated from the first dentary tooth by a gap that is between one and two tooth positions long.

The posteroventral process extends further posteriorly than the posterolateral processes (Fig. 14B: vppd), and the posterior third is distinctly bifurcated as in the neornithischians Changchunsaurus, Haya, and Talenkauen (Novas, Cambiaso & Ambrosio, 2004; Jin et al., 2010; Makovicky et al., 2011), basal ceratopsians (e.g., Archaeoceratops, Liaoceratops, Yinlong: Xu et al., 2002; You & Dodson, 2003; Xu et al., 2006), and some iguanodontians (e.g., Zalmoxes, Dryosaurus: Weishampel et al., 2003; Norman, 2004). The oral margin of the predentary is smooth and relatively straight, though the anterior tip is slightly dorsally projected (Fig. 14A), though not to the extent seen in some basal ceratopsians (e.g., Ajkaceratops, Liaoceratops: Xu et al., 2002; Osi, Butler & Weishampel, 2010). The anterior-most portion of the oral margin consists of a sharp ridge. This ridge flattens out and becomes mediolaterally wider posteriorly to form a triangular shaped, dorsally flattened surface that is broadly concave. The posterior-most premaxillary tooth ends just anterior to the posterior end of the oral margin of the predentary, and elongation of the premaxilla prevents the anterior maxillary teeth from occluding with the predentary. The articulation surface for the dentary consists of a broad, ‘u-shaped’ sulcus that extends from the posterior-most tip of the posterolateral processes down to the posterior-most tip of the posteroventral process (Fig. 14A).

Dentary

The dentary forms the majority of the mandibular ramus and is the only tooth bearing element of the lower jaw. There are twenty tooth positions within the dentary. The thyreophoran dinosaur Scutellosaurus and the neornithischians Agilisaurus and Hexinlusaurus all possess at least 18 dentary tooth positions (Colbert, 1981; He & Cai, 1984; Peng, 1992). The tooth row is not sinuous like it is derived thyreophorans (Norman, Witmer & Weishampel, 2004b; Butler, Upchurch & Norman, 2008), but the anterior portion of the dorsal surface slopes anteroventrally, causing the anterior three tooth positions to be offset ventrally below the rest of the dentary tooth row and the crowns are angled anterodorsally (Fig. 14C), a condition not seen in any other neornithischian taxon (Norman et al., 2004). The anterior-most tip of the dentary is spout-shaped (Fig. 14D), as in all ornithischians except Eocursor and heterodontosaurids (Butler, Smith & Norman, 2007; Butler, Upchurch & Norman, 2008), and is positioned nearly level with the ventral margin of the dentary.

The dorsal and ventral margins of the dentary converge anteriorly (Fig. 14C), which is the primitive condition within Ornithischia. As in all ornithischians, the posterodorsal portion of the dentary forms the anterior portion of the coronoid process, contacting the coronoid medially and the surangular posteriorly. The posterior-most portion of the tooth row is situated medial to the rising coronoid process, as in the neornithischian taxa Changchunsaurus and Jeholosaurus, and in most iguanodontians (Jin et al., 2010; Norman, 2004; Barrett & Han, 2009). The lateral surface of the dentary is convex dorsoventrally with a pronounced ridge present that begins posteriorly at the base of the dentary contribution to the coronoid process, extends slightly anteroventrally to mid-length, then arcs anterodorsally, gradually becoming less pronounced and terminating near the first dentary tooth position (Fig. 14C). The lateral surface of the anterior third of the dentary is covered by numerous, irregularly distributed foramina, while posteriorly a few foramina are present in a row just dorsal to the lateral ridge.

The medial surface of the posterior end of the dentary is dorsoventrally concave and overlapped the lateral surfaces of the angular, coronoid, and surangular. The ventral surface is anteroposteriorly concave and mediolaterally convex. The medial surface of the dentary is convex both anteroposteriorly and dorsoventrally (Fig. 14D), as in all neornithischians except Othnielosaurus and Zalmoxes (C Boyd, pers. obs., 2011, Weishampel et al., 2003; Norman et al., 2004; Godefroit, Codrea & Weishampel, 2009). There is a row of replacement foramina positioned ventromedial to the tooth row, as in all genasaurians and the heterodontosaurid Fruitadens (Norman, Witmer & Weishampel, 2004a; Butler et al., 2012). The Meckelian groove is situated near the ventral margin. It begins near the anterior end as a shallow groove that is dorsoventrally broader than mediolaterally deep. As the groove extends posteriorly, it becomes slightly taller and substantially deeper, angling dorsolaterally into the dentary lateral to the roots of the dentary teeth. Near the posterior end, the anterior portions of the angular, prearticular, and surangular insert into this groove. The splenial overlaps the medial surface of the dentary, with its thickened, slightly ventrolaterally curved ventral margin sitting over the Meckelian groove.

Coronoid

The coronoid is composed of three processes (Figs. 16H and 16I), which differs from the strap-like coronoid present in Lesothosaurus, the thyreophoran Scelidosaurus, and the heterodontosaurid Lycorhinus (Sereno, 1991). The lobate dorsal process of the coronoid is positioned medial to the coronoid rise of the dentary and contacted the anterodorsal margin of the surangular (Figs. 16A and 16B). The morphology and position of the dorsal process is similar to that of the neornithischians Changchunsaurus (Jin et al., 2010) and Hypsilophodon (Galton, 1974a) and the basal ceratopsian Psittacosaurus (Sereno, 1987). A short ventral process was overlapped medially by the splenial, and was separated from the anterior process by a shallow sulcus (Figs. 16H and 16I). This morphology differs from that seen in Hypsilophodon, in which the coronoid is triangular in medial view, lacking an anteroventral sulcus between the ventral and anterior projections (Galton, 1974a). The posteroventral portion of the coronoid is obscured by the splenial in Changchunsaurus (Jin et al., 2010). The anterior process of the coronoid tapered anteriorly (Figs. 16H and 16I) and was relatively short compared to the elongate coronoids of Changchunsaurus (Jin et al., 2010) and Lesothosaurus (Sereno, 1991), but similar in length and shape to the coronoid of Hypsilophodon (Galton, 1974b), extending medial to the posterior five dentary tooth positions.

Surangular

The anterior portion of the surangular is slightly medially deflected in dorsal view (Fig. 15G), mediolaterally thin, and was situated medial to the dentary (Fig. 15A). The anterior two-thirds of the ventral margin angles medioventrally and was overlapped laterally by the lateral wall of the angular (Fig. 15H). The dorsal margin of the surangular is convex in lateral view (Fig. 15E), which is the basal ornithischian condition. The dorsal-most portion of the surangular is triangular in lateral view, is convex medially, and forms the posterodorsal portion of the coronoid eminence (Fig. 16A). The anterodorsal margin contacted the dentary along its lateral half and the coronoid along its medial half.

The lateral surface of the surangular is flattened and oriented slightly posteriorly. Two distinct foramina are present on the lateral surface of the left surangular just posterior to the contact with the dentary (Fig. 15E: sf1 and sf2), but only a single foramen is present in the same area on the right surangular (Fig. 15A). The lateral foramina on the left surangular converge and exit the medial surface of the surangular through a single foramen (Fig. 15F: sf). Thus, the two foramina on the left surangular and the single foramen on the right surangular represent the surangular foramen that is also present in the neornithischians Changchunsaurus, Gasparinisaura, Hypsilophodon, Jeholosaurus, Orodromeus, and Oryctodromeus, the basal ceratopsian Yinlong, and most basal iguanodontians (Galton, 1974a; Coria & Salgado, 1996; Scheetz, 1999; Norman, 2004; Xu et al., 2006; Barrett & Han, 2009; Jin et al., 2010; C Boyd, pers. obs., 2011). On the left dentary of the holotype of Thescelosaurus edmontonensis (CMN 8537; now referred to Thescelosaurus sp. (Boyd et al., 2009)) the surangular foramen consists of a single opening laterally, but bifurcates into two foramina by the time it exits the medial surface of the surangular (Galton, 1974b: Figs. 1E and 1G). These observations indicate there is considerable variation in the morphology of the surangular foramen in Thescelosaurus, even within a single specimen.

There is a distinct, dorsolaterally directed, ‘finger-like’ process on the lateral surface of the surangular next to the glenoid (Figs. 15E–15H), similar to the structure seen in the basal iguanodontians Tenontosaurus tilletti and Zalmoxes robustus (Weishampel et al., 2003; Norman, 2004). However, the dorsolateral process present in NCSM 15728 is dorsoventrally taller than anteroposteriorly wide (Fig. 15E), while the reverse condition is present in Tenontosaurus tilletti and Zalmoxes robustus (Weishampel et al., 2003; Norman, 2004). In several neornithischian taxa a low boss is present in this same region (e.g., Changchunsaurus, Haya, Hypsilophodon, Orodromeus, Zephyrosaurus: Galton, 1974a; Galton, 1997; Scheetz, 1999; Jin et al., 2010; Makovicky et al., 2011), though there is no boss or process present in the basal ornithischian Lesothosaurus (Sereno, 1991). Just anterior to the base of the dorsolateral process a single foramen is present on the left surangular (Fig. 15E: lpf), while two foramina are present on the right surangular, with the second foramen positioned on the dorsolateral process (Fig. 15A). Both foramina on the right surangular connect in the inside of the surangular and exit through a single, anteriorly facing foramen on the medial surface of the surangular just anterior to the medial process of the surangular, which is the same location of the exit of the single foramen on the left surangular. A foramen is also present in this area in other neornithischian taxa (e.g., Changchunsaurus, Haya, Hypsilophodon: Galton, 1974a; Jin et al., 2010; Makovicky et al., 2011).

The posterior portion of the surangular is anteroposteriorly elongate and dorsoventrally narrow, forming the lateral and part of the ventral cup for the articular (Fig. 15E: rp). The lateral surface of the posterior margin of the retroarticular process is covered by a series of anteroposteriorly oriented grooves. The medial surface of the surangular is dorsoventrally and mediolaterally concave. Directly medial to the base of the dorsolateral process, a dorsoventrally flattened process projects medially (Figs. 15F and 15G), contacting a small process on the prearticular and forming a distinct foramen posterior to the articular (Fig. 16B). This process is also present in the holotype of Thescelosaurus edmontonensis (Galton, 1997). A similar process is not present in Hypsilophodon (Galton, 1974a), but the morphology of the medial surface of the surangular is poorly known in other neornithischian taxa.

Angular

The angular forms the posteroventral portion of the mandible (Figs. 15A, 16A and 16C). The lateral wing of the angular extends dorsally much higher than the medial wing (Fig. 15B versus Fig. 15C). The anterior portion of the lateral wing is triangular-shaped and positioned medial to the dentary (Fig. 15B). Much of the medial surface of the lateral wing overlapped the ventrolateral surface of the surangular. The majority of the exposed lateral surface forms a shallow fossa that extends dorsally onto the surangular (Figs. 15A, 15B and 15E). The ventral margin of the angular is broadly convex anteroposteriorly and rounded mediolaterally (Figs. 15B and 15D). The ventral surface is mediolaterally widest posteriorly where it formed the ventral portion of the retroarticular process (Fig. 15D). The short medial wing has a complex contact with the prearticular (Figs. 15C and 15D). The posterior-most portion of the medial wall overlapped the ventral surface of the prearticular medially, but the majority of the medial wing of the angular was positioned lateral to a short ventral flange of the prearticular, resulting in the presence of a narrow, dorsomedially facing contact surface on the medial wing of the angular (Fig. 15C: paa). Ventral to this contact surface for the prearticular, a second, dorsoventrally narrow contact surface is present for the posterior process of the splenial (Figs. 15C and 15D: sas).

Splenial

The splenial is a thin, ‘plate-like’ bone positioned along the posteromedial portion of the mandible (Fig. 16A). The majority of the splenial is mediolaterally thin, except for the ventral margin which is thickened where it overlapped the Meckelian groove of the dentary (Fig. 16E). The medial surface is dorsoventrally concave. The anterior two-thirds is triangular (Fig. 16D) with the narrow anterior tip positioned near the Meckelian groove along the ventral margin of the dentary and the maximum dorsoventral height positioned close to the level of the posterior-most tooth position of the dentary. No foramen is present near the anterior tip of the splenial, unlike in the neornithischians Changchunsaurus and Haya (Jin et al., 2010; Makovicky et al., 2011) and in saurischian dinosaurs (Rauhut, 2003).

The posterior third of the splenial consist of a dorsoventrally narrow posterior process that contacted the prearticular and angular medially (Fig. 16A). This posterior process is not bifurcate, unlike the neornithischian Changchunsaurus and the marginocephalians Archaeoceratops (Jin et al., 2010). The splenial is not visible in lateral view along the ventral margin of the dentary in this specimen (Figs. 1 and 2), but because both splenials are slightly displaced from life position this may not have been the natural condition.

Prearticular

The prearticular is a mediolaterally flatted bone that forms the posteromedial-most portion of the mandible (Figs. 16A and 16C). The posterior end is laterally concave where it rested against the articular, forming the medial surface of the retroarticular process. The anterior portion of the prearticular consists of two processes (Figs. 16D and 16E). The anterodorsal process is dorsoventrally tall and mediolaterally thin, angled anterodorsally, and was situated lateral to the splenial. Unlike the prearticular of Changchunsaurus, the anterior end is not twisted to face ventromedially (Jin et al., 2010). The anteroventral process was dorsoventrally narrow and was situated between the angular and the splenial. A prominent ridge is present on the lateral surface of the anteroventral process that extends posterior about two-thirds of the length of the prearticular (Fig. 16G: ra). The surface ventral to this ridge demarcates the contact for the angular. Posteriorly, this contact surface rotates from ventrally facing to medially facing, where the posterior-most portion of the angular overlapped the lateral surface of the prearticular (Fig. 16F: aa).

This complex contact with the angular is atypical for neornithischians, where the angular generally overlaps the ventral edge of the prearticular medially (e.g., Hypsilophodon: Galton, 1974a). There is no evidence of the narrow slit in the posterior portion of the prearticular noted in Hypsilophodon (Galton, 1974a). At approximately midlength along the dorsal margin a short, laterally projecting process is present that contacted a corresponding medially directed process on the surangular (Figs. 16F and 16G: lpp), creating a foramen anterior to the articular (Fig. 16B).

Articular

The articular is roughly rectangular in both lateral (Fig. 16J) and dorsal views (Fig. 16K), being slightly anteroposteriorly longer than dorsoventrally tall, unlike the triangular articular of Hypsilophodon or the elliptical articular of Agilisaurus (Galton, 1974a; Peng, 1992). The articular is positioned within a distinct cup formed by the prearticular medially, the angular ventrally, and the surangular laterally, which is generally the case in neornithischians (e.g., Hypsilophodon, Orodromeus: Galton, 1974a; Scheetz, 1999). The anterodorsal surface is mediolaterally convex, for articulation with the distal condyles of the quadrate (Fig. 16K).

Accessory ossifications

Supraorbital

The supraorbital bar is composed of two elements (Figs. 17A and 17B): the supraorbital, often referred to as a palpebral (e.g., Barrett, Butler & Knoll, 2005; Barrett & Han, 2009; Jin et al., 2010; Maidment & Porro, 2010; Makovicky et al., 2011); and, an accessory supraorbital (= postpalpebral of Makovicky et al. (2011) and the supraorbital of Barrett, Butler & Knoll (2005)). An accessory supraorbital is also present in the neornithischian taxa Agilisaurus and Haya (Barrett, Butler & Knoll, 2005; Makovicky et al., 2011) and multiple supraorbitals (up to 3) are present in some derived thyreophorans and pachycephalosaurians (Maryańska, Chapman & Weishampel, 2004; Norman, Witmer & Weishampel, 2004b; Maidment & Porro, 2010). The supraorbitals are free of the orbital margin and project across the orbit (Figs. 1 and 3), unlike in derived thyreophorans and pachycephalosaurids where it is incorporated into the orbital margin (Maryańska, Chapman & Weishampel, 2004; Norman, Witmer & Weishampel, 2004b). The supraorbital bar transverses the entire width of the orbit (Figs. 1 and 3), as in the neornithischians Agilisaurus and possibly Haya (the two supraorbitals may not contact each other in this taxon; Makovicky et al., 2011) and the basal iguanodontian Dryosaurus altus (Galton, 1983). A supraorbital bar that transverses the entire orbit was proposed to be a local autapomorphy of Agilisaurus (Barrett, Butler & Knoll, 2005), but this feature is more widespread among neornithischians than previously suspected.

The anterior facet is medially concave and rugose where it formed a loose articulation against the roughened surfaces on the prefrontal and the lacrimal at the anterodorsal corner of the orbit (Fig. 17C). The supraorbital articulation also spans the prefrontal and lacrimal in the heterodontosaurid Heterodontosaurus (Crompton & Charig, 1962), the neornithischians Agilisaurus and Orodromeus (Peng, 1992; Scheetz, 1999), and in some ceratopsians (e.g., Archaeoceratops: You & Dodson, 2003). The supraorbital formed the anterior two-thirds of the supraorbital bar. The anterior facet is medially concave with a prominent dorsomedially directed process extending from the posterodorsal margin that overlapped the posterior surface of the prefrontal, giving the proximal end a triangular outline in proximal view (Fig. 16C). The dorsal margin of the anterior facet is lined with a series of small rugose projections. The rod-shaped posterior process of the supraorbital is posterodorsally oriented in lateral view (Figs. 1 and 17B), the dorsal and ventral margins converge posteriorly, and the distal tip curves to face nearly directly posterior. In dorsal view the posterior process angles posterolaterally along most of its length, is mediolaterally broad with a slightly convex surface, and remains a nearly constant thickness until the distal end, which curves posteriorly and tapers to a blunt point (Fig. 17A). The distal tip is covered with a series of anteroposteriorly oriented ridges they may have facilitated a soft tissue connection to the accessory supraorbital. The entire surface of the supraorbital is covered with a series of anteroposteriorly oriented striations (Fig. 17B). The dorsomedial margin is covered with a series of rugose projections (Fig. 17B), possibly for connection to associated soft tissues (Scheetz, 1999).

The accessory supraorbital forms the posterior third of the supraorbital bar and is approximately half the length of the supraorbital (Figs. 17A and 17B). The accessory supraorbital is proportionally larger than those in the neornithischians Agilisaurus and Haya (Peng, 1992; Makovicky et al., 2011). The medial surface is flattened both anteroposteriorly and dorsoventrally, while the lateral surface is convex in both directions (Fig. 17A). In lateral view the dorsal margin is concave and the ventral margin is convex. The anterior third of the accessory supraorbital is oriented anterodorsally, is dorsoventrally narrower than the posterior two-thirds, and is covered laterally with a series of fine, anteroposteriorly oriented ridges (Fig. 17B). The posterior two-thirds is oriented posteriorly and the margins are rugose where it overlapped a flattened facet on the lateral surface of the postorbital (Figs. 3 and 17A).

Hyoid

The ceratobranchials were preserved near the posteroventral corner of the mandible. They were subsequently separated from the specimen and are now isolated elements. Each ceratobranchial consists of an elongate, ‘rod-shaped’ bone that is strongly curved so that it is dorsally concave and ventrally convex in lateral view (Figs. 17D and 17E). The anterior half was apparently oriented near the ventral margin of the posterior portion of the mandible, while the posterior half curved dorsally around the posterior end of the mandible, closely matching the general morphology and position of the ceratobranchial in iguanodontian ornithischians and basal sauropods (Norman, 2004; Upchurch, Barrett & Dodson, 2004). The anterior portion of the ceratobranchial is oriented roughly horizontal and the anterior end is slightly dorsoventrally expanded and ovate in cross-section. The posterior portion is oriented posterodorsally at an angle of approximately forty-five degrees from the anterior portion. The posterior portion tapers dorsoventrally and becomes progressively mediolaterally flattened towards the posterodorsal tip (Figs. 17D and 17E). The morphology of the ceratobranchial differs from those preserved in the neornithischians Changchunsaurus, Jeholosaurus, Hypsilophodon, and Parksosaurus (Galton, 1973; Galton, 1974a; Barrett & Han, 2009; Jin et al., 2010), which are relatively straight and do not show the strong curvature present in NCSM 15728. The ceratobranchials may be curved in the taxon Agilisaurus (Peng, 1992: Fig. 1) and are preserved in a more anterior position than in NCSM 15728, though it is uncertain if they are distorted or if they were displaced from their original position.

Sclerotic plates

Isolated sclerotic plates are present, randomly distributed throughout the orbit. These plates are extremely mediolaterally thin and fragile. The best exposed of these is preserved lying on the dorsal surface of the parasphenoid (Fig. 2: sp), though it is distorted from being pressed against the underlying bone. As a result, not much can be said regarding the morphology of these plates or of the morphology of the sclerotic ring.

Dentition

Premaxillary dentition

Six teeth are present in each premaxilla, as in the basal ornithischian Lesothosaurus (Sereno, 1991), the basal thyreophoran Scutellosaurus (Colbert, 1981), and the neornithischian Jeholosaurus (Barrett & Han, 2009). However, it appears that the number of premaxillary teeth increases during ontogeny based on examination of multiple specimens of the basal ornithischian taxa Jeholosaurus (e.g., PKUP V 1064: C Boyd, pers. obs., 2011) and Thescelosaurus (C Boyd, pers. obs., 2011). Thus, the lower tooth counts observed in some other neornithischian taxa may not reflect the number present in mature individuals of all of those taxa.

The premaxillary crowns are slightly mediolaterally compressed and slightly constricted at their bases (Fig. 18A). The bluntly pointed distal tips of the crowns are recurved posteriorly. Serrations are absent on both the anterior and posterior margins just as in the neornithischians Changchunsaurus, Haya, and Jeholosaurus (Barrett & Han, 2009; Jin et al., 2010; Makovicky et al., 2011), but weakly developed carinae are present that are more pronounced on the anterior margins. On some premaxillary crowns (e.g., Fig. 18A: pmt6) a dorsoventrally oriented groove is present adjacent to the carina, which is also seen in the basal ornithischian Lesothosaurus (Sereno, 1991) and the neornithischian Jeholosaurus (Barrett & Han, 2009). The surfaces of the premaxillary crowns are ornamented by numerous fine ridges that extend from the distal tip to the base of the crown. Similar ornamentation is present in Hypsilophodon (Galton, 1974a), but is absent in Changchunsaurus, Jeholosaurus, and Zephyrosaurus (Sues, 1980; Barrett & Han, 2009; Jin et al., 2010). In NCSM 15728, these ridges are less prominent in teeth that display a higher degree of wear (Fig. 18A: pmt3). Enamel is evenly distributed on all sides of the crowns. The premaxillary tooth crowns of NCSM 15728 differ from those of most heterodontosaurids (except Friutadens: Butler et al., 2010; Butler et al., 2012), in which the crowns are straight, subcylindrical, and unconstricted at their base (Butler, Upchurch & Norman, 2008).

The roots of the premaxillary teeth are elliptical in Jeholosaurus (Barrett & Han, 2009), but round in Hypsilophodon and Zephyrosaurus (Galton, 1974a; Sues, 1980). In NCSM 15728, the shape of the premaxillary tooth roots vary based on tooth position, with the more anteriorly positioned teeth possessing roots that are elliptical in cross section (mediolaterally compressed), while the posterior-most teeth possess roots that are roughly circular in cross-section. The roots of premaxillary teeth four through six are oriented dorsomedially away from the crowns; however, the roots of the anterior three premaxillary teeth progressively angle more posteriorly as well, with the root for the first premaxillary tooth oriented posteriorly at roughly a forty-five degree angle from the long axis of the body of the premaxilla. This same pattern is observed in the partial premaxillae of the holotype of Bugenasaura (SDSM 7210), which is now referred to Thescelosaurus (Boyd et al., 2009). Medially oriented wear facets were reported on isolated premaxillary teeth referred to Thescelosaurus (Galton, 1974b). Medially facing wear facets are also reported for the neornithischian Zephyrosaurus (Sues, 1980). Alternatively, the wear facets on the premaxillary teeth of NCSM 15728 are on the distal tips of the crowns and progressive wear decreases the height of the crown. In Jeholosaurus, both patterns of wear are observed (Barrett & Han, 2009). Replacement teeth are present in some of the premaxillary alveoli, with a single replacement tooth positioned medial to the root of the erupted tooth. No set pattern of tooth replacement is readily apparent from examination of the CT data.

Maxillary dentition

The maxillary tooth row is inset from the lateral margin of the maxilla and overhung by a prominent, anteroposteriorly oriented ridge on the maxilla (Figs. 1 and 2). In Lesothosaurus and Scutellosaurus the maxillary teeth are only modestly inset from the lateral margin (Colbert, 1981; Sereno, 1991). Twenty teeth are present in each maxilla, more than in any other neornithischian (Norman et al., 2004). The anterior end of the maxillary tooth row is more posteriorly positioned than the anterior end of the dentary tooth row (Figs. 1 and 2). As a result, the first maxillary crown occludes with anterior margin of the fourth dentary crown and possibly with the posterior margin of the third dentary crown (Fig. 18B). Posteriorly the tooth row wraps around the posterior end of the maxilla, causing the posterior-most maxillary crowns to be oriented posteroventrally instead of directly ventrally (Fig. 18C). This does not appear to be a result of distortion of the specimen because it occurs on both sides of the specimen, and in the CT data the posterior ends of the maxillae appear undamaged. Instead, this may result from the high number of teeth present in the maxillary and dentary tooth rows compared to other neornithischians and the fact that the posterior end of the dentary tooth row extends medial to the rising coronoid process and, as a result, the posterior-most dentary teeth are slightly more dorsally positioned than the anterior portion of the dentary tooth row.

Unworn maxillary crowns are roughly triangular in shape in lateral view and their dorsoventral height is approximately equal to their anteroposterior width (Figs. 18B and 18C), as in the heterodontosaurid Echinodon (Galton, 2007), the basal thyreophoran Scutellosaurus (Colbert, 1981), the neornithischians Changchunsaurus, Jeholosaurus, Orodromeus, Othnielosaurus, and Zephyrosaurus (Sues, 1980; Scheetz, 1999; Barrett & Han, 2009; Jin et al., 2010; C Boyd, pers. obs., 2011), and the basal ceratopsian Yinlong (Xu et al., 2006). The maxillary teeth are arranged en echelon, with the posterior portion of each crown positioned lateral to the anterior portion of the proceeding crown (Figs. 18B and 18C). The roots of the maxillary teeth are spaced apart from each other (Fig. 18C), unlike in more derived ornithopod and ceratopsian dinosaurs where the roots of adjacent teeth tightly contact each other (Norman, 2004; You & Dodson, 2004). There is a distinct constriction, or neck, present at the base of the crown, as in all neornithischians except Hypsilophodon and Jeholosaurus (Galton, 1974a; Barrett & Han, 2009). Distal to this constriction, a distinct cingulum is present at the base of the crown, as in all neornithischians (Norman et al., 2004). The medial surfaces of the maxillary crowns are convex, as in the neornithischians Hypsilophodon, Leaellynasaura, and Zephyrosaurus (Galton, 1974a; Sues, 1980; Rich & Vickers-Rich, 1999) and in iguanodontians (Norman, 2004). The distribution of enamel on the maxillary crowns is rather symmetrical, as in all neornithischians except Hypsilophodon (Galton, 1974a) and in most heterodontosaurids except Abrictosaurus and Heterodontosaurus (Butler, Upchurch & Norman, 2008).

Marginal denticles are present on the maxillary crowns and extend to near the base of the crown, unlike in all heterodontosaurids and the basal ceratopsian Chaoyangsaurus (Zhao, Cheng & Xu, 1999). The marginal denticles are confluent with ridges that extend to the base of the crown, unlike in an unnamed taxon from the Kaiparowits Formation of Utah (Boyd, 2012; Gates et al., 2013), though in this latter taxon the absence of these ridges may reflect the early ontogenetic stage of the specimen that preserves the partial maxilla. A prominent, primary ridge on the lateral surface near the apex of the crown is absent in NCSM 15728, unlike in the heterodontosaurid Heterodontosaurus (Crompton & Charig, 1962), the neornithischian Talenkauen (Novas, Cambiaso & Ambrosio, 2004), some basal ceratopsians (e.g., Archaeoceratops: You & Dodson, 2003), and most basal iguanodontians except Rhabdodon, Tenontosaurus, and Zalmoxes (Norman, 2004; Weishampel et al., 2003). The presence of ridges on the lateral surface of the maxillary crowns that form two converging crescentic patterns was proposed to be an autapomorphy of Thescelosaurus by Galton (1997). However, the presence of this feature is variable in NCSM 15728, with some teeth displaying this feature (e.g., Fig. 18B: mt2) while adjacent teeth display nearly vertical ridges (e.g., Fig. 18B: mt3). Thus, this character was dismissed as an autapomorphy of Thescelosaurus in the recent review of the taxon by Boyd et al. (2009).

The maxillary tooth roots are dorsoventrally straight, as in all basal ornithischians. In general, the maxillary teeth do not form a continuous occlusion surface, with each maxillary crown offset in between two dentary crowns, creating distinct anterolingual and posterolingual wear surfaces on the maxillary crowns. However, on the posterior maxillary teeth a single, roughly horizontal wear facet is present on each crown that closely matches the height of the wear facets on the adjacent teeth, creating a nearly continuous occlusion surface (Fig. 18C). There is a maximum of one replacement tooth present in each alveolus, with the newly forming tooth positioned lingual to the erupted tooth.

Dentary dentition

The dentary teeth are poorly exposed in NCSM 15728. On the right side of the skull, only the anterior three dentary teeth are visible, and the maxillary dentition obscures the more posterior dentary crowns (Fig. 1). On the left side of the skull the dentary and maxilla are slightly separated, allowing the lateral surfaces of the anterior nine dentary crowns to be seen, and parts of the next three crowns, but the posterior eight crowns are entirely obscured by the overlapping maxillary dentition (Figs. 2 and 19). CT data was used to gather additional information regarding the morphology of the dentary teeth, but the resolution of the scans is insufficient to fully elucidate their morphology. The dentary tooth row is inset from the lateral margin of the dentary, and a prominent anteroposteriorly oriented ridge present on the dentary ventral to the tooth row (Figs. 1 and 2). Twenty teeth are present in the dentary. The neornithischian taxa Agilisaurus and Hexinlusaurus also possess twenty dentary teeth (He & Cai, 1984; Peng, 1992), but in these taxa the number of dentary teeth is greater than the number of maxillary teeth, while NCSM 15728 possesses an equal number of dentary and maxillary teeth. The roots of the dentary teeth are dorsoventrally straight, unlike the dorsoventrally curved dentary tooth roots seen in the neornithischians Hypsilophodon and Parksosaurus (Galton, 1974a; C Boyd, pers. obs., 2011) and in iguanodontians (Norman, 2004).

The anterior-most dentary tooth is more anteriorly positioned than the anterior-most maxillary tooth (Fig. 19). As a result, the anterior two, and possibly also the third, dentary teeth do not occlude with the maxillary dentition; rather, they are situated ventral to the premaxillary-maxillary diastema. In Agilisaurus, the anterior three dentary teeth extend anterior beyond the maxillary tooth row, but they occlude with the premaxillary teeth owing to the lack of a premaxillary-maxillary diastema in that taxon (Barrett, Butler & Knoll, 2005). None of these anterior dentary teeth match the morphology of the enlarged, anteriorly positioned caniniform tooth present in the heterodontosaurids Fruitadens, Heterodontosaurus, Lycorhinus, and Tianyulong (Crompton & Charig, 1962; Hopson, 1975; Zheng et al., 2009; Butler et al., 2010; Norman et al., 2011; Sereno, 2012)). The anterior-most dentary tooth is not reduced relative to the other dentary teeth, as is the first dentary tooth in Agilisaurus (Barrett, Butler & Knoll, 2005). The anterior two dentary teeth are slightly more enlarged than dentary teeth 3–5 and they are anteroposteriorly narrower than the other dentary crowns (Fig. 19). The posterior margins of the first three dentary crowns are slightly concave, but the crowns are not recurved like the anterior three dentary teeth in Agilisaurus (Barrett, Butler & Knoll, 2005). The anterior three dentary teeth bear marginal denticles and confluent ridges, but they are reduced in number and prominence compared to the more posterior dentary crowns.

The remainder of the dentary crowns are roughly ‘triangular-shaped’ in lateral view and their dorsoventral height is less than 150% of the anteroposterior width of the crown. In all of the dentary teeth, a distinct constriction, or neck, is present between the base of the dentary crown and its corresponding root. A cingulum is present along the base of the crown, as in all neornithischian dinosaurs (Norman et al., 2004). Marginal denticles are present on both the anterior and posterior edges of the dentary crowns, and these denticles are confluent with ridges that extend to the base of the crown, as in the heterodontosaurids Heterodontosaurus and Tianyulong (Crompton & Charig, 1962; Zheng et al., 2009), the neornithischians Haya, Hypsilophodon, Jeholosaurus, Othnielosaurus, Parksosaurus, and Talenkauen (Galton, 1973; Galton, 1974a; Galton, 2007; Novas, Cambiaso & Ambrosio, 2004; Barrett & Han, 2009; Makovicky et al., 2011), some basal ceratopsians (e.g., Archaeoceratops and Liaoceratops: Xu et al., 2002; You & Dodson, 2003), and basal iguanodontians (Norman, 2004). These ridges are present on both the medial and lateral surfaces of the dentary crowns, unlike in heterodontosaurid Heterodontosaurus (Norman et al., 2004), the neornithischian Hypsilophodon (Galton, 1974a), some basal ceratopsians (e.g., Liaoceratops: Xu et al., 2002), and most basal iguanodontians (Norman, 2004) where ridges are limited to the medial side of the crown. The apex of the dentary crowns is centrally to slightly anteriorly positioned on the crown, unlike in the basal ornithischian Lesothosaurus (Sereno, 1991), the basal marginocephalian Wannanosaurus (Butler & Zhao, 2009), and dryomorph iguanodontians (Norman, 2004) where the apex is positioned posteriorly on the crown. Well-developed wear facets are present on the anterolateral and posterolateral surfaces of the anterior dentary crowns, indicating that each dentary crown occluded with two maxillary teeth and that a continuous occlusion surface was not present on the anterior dentary teeth (Fig. 19). It cannot be determined if the wear pattern on the posterior dentary teeth resembled that seen on the anterior dentary crowns or if a nearly continuous occlusion surface was developed as seen in the posterior maxillary teeth. As in the maxillary dentition, a maximum of one replacement tooth is present in each alveolus positioned lingual to the erupted tooth.

Discussion

The conflicting cranial character data of Thescelosaurus neglectus

Thescelosaurus neglectus is an unusual neornithischian. Its large body size (>four meters: Fisher et al., 2000; C Boyd, pers. obs., 2011) is in sharp contrast to the general body size range displayed by most other basal neornithischians and basal ornithopods (∼1–2 m: Norman et al., 2004). Additionally, T. neglectus displays an eclectic set of plesiomorphic and apomorphic characters that complicate attempts to resolve its systematic placement within Neornithischia and to identify its sister taxon. The detailed cranial description of T. neglectus presented above highlights an even more discordant mixture of plesiomorphic and apomorphic characters in this taxon than was previously recognized. The systematic relationships of T. neglectus will be thoroughly analyzed elsewhere using these new character data (C Boyd, 2012, unpublished data), but a detailed discussion of plesiomorphic and apomorphic traits displayed in the skull of this taxon in light of the new character evidence presented herein and new character data from other neornithischian taxa is pertinent to the current discussion.

The dentition of T. neglectus displays a suite of characters unique to this taxon. The presence of six premaxillary teeth in Thescelosaurus neglectus and Jeholosaurus would seem to indicate independent reversals in those taxa to the plesiomorphic condition based on the presence of six premaxillary teeth in the basal ornithischian Lesothosaurus and the basal thyreophoran Scutellosaurus (Colbert, 1981; Sereno, 1991). However, the number of premaxillary teeth is found to vary during ontogeny in Jeholosaurus (e.g., PKUP V 1064: C Boyd, pers. obs., 2011) and Thescelosaurus (e.g., SDSM 7210). Based on these new data, the presence of six premaxillary teeth may be more widespread among neornithischian taxa, with the full distribution of this character clouded by the fact that some taxa are known largely from ontogenetically immature specimens (e.g., Orodromeus: Scheetz, 1999). Thescelosaurus neglectus also shares with Lesothosaurus (and Jeholosaurus) the presence of a dorsoventrally oriented groove adjacent to the carina on the premaxillary crowns (Sereno, 1991; Barrett & Han, 2009). Alternatively, T. neglectus and Hypsilophodon are unique in possessing fine, dorsoventrally oriented ridges on the premaxillary crowns (Galton, 1974a), and the premaxillary crowns of T. neglectus, Changchunsaurus, Haya, and Jeholosaurus differ from other basal ornithischians in lacking serrations (Barrett & Han, 2009; Jin et al., 2010; Makovicky et al., 2011).

The maxillae and dentaries both contain a maximum of twenty tooth positions, a condition that more closely resembles the basal genosaurian condition, and deviates from a general trend in neornithischians and basal iguanodontians of reducing the number of tooth positions in both the maxillae and dentaries (Norman, 2004; Norman et al., 2004). The roots of the dentary and maxillary teeth are dorsoventrally straight and the enamel is symmetrically distributed on the dentary and maxillary crowns in T. neglectus, which are ornithischian symplesiomorphies (Scheetz, 1999; Butler, 2005; Butler, Upchurch & Norman, 2008). A distinct constriction, or neck, is present at the base of each crown, and just dorsal to that constriction a distinct cingulum is present, both of which are plesiomorphic for Neornithischia (Weishampel & Heinrich, 1992; Scheetz, 1999). The presence of both a convex medial surface on the maxillary crowns and the presence of ridges on the surfaces of the maxillary and dentary crowns that extend from the marginal denticles to the base of the crown are shared with basal iguanodontian taxa, along with some other neornithischians (Scheetz, 1999; Weishampel et al., 2003). Additionally, the posterior end of the dentary tooth row extends medial to the rising coronoid process, a feature seen in Changchunsaurus, Jeholosaurus, and most iguanodontians (Weishampel et al., 2003; Barrett & Han, 2009; Jin et al., 2010).

The lower jaw of T. neglectus displays two apomorphic features generally seen in basal iguanodontians. The posteroventral process of the predentary is bifurcated in T. neglectus, which is a character commonly associated with basal iguanodontians; however, the neornithischians Changchunsaurus and Haya (Jin et al., 2010; Makovicky et al., 2011) and some basal ceratopsians (You & Dodson, 2003; You & Dodson, 2004) also display this feature, suggesting the character state has a much wider distribution than previously assumed. The surangular of T. neglectus bears a dorsolateral process near the lateral margin of the glenoid, similar in appearance to the dorsally projecting lip positioned lateral to the glenoid in the basal iguanodontians Tenontosaurus tilletti and Zalmoxes robustus (Weishampel et al., 2003; Norman, 2004). Some other neornithischian taxa possess small bosses near the lateral margin of the glenoid (e.g., Changchunsaurus; Jin et al., 2010), but none of these are as well developed as seen in T. neglectus.

The morphology of the braincase in T. neglectus is relatively derived with respect to basal ornithischians, and most closely resembles that of Dysalotosaurus (Galton, 1989), though it may lack an ossified orbitosphenoid. The basioccipital bears a ventral midline keel and an arched floor of the braincase, both of which are plesiomorphic for Neornithischia (Scheetz, 1999) and present in some basal iguanodontians (e.g., Dysalotosaurus: Scheetz, 1999; Norman, 2004). The anteroposterior length of the basioccipital is less than that of the basisphenoid (not including parasphenoid) in T. neglectus, T. assiniboiensis, Jeholosaurus, and some basal iguanodontians (Norman, 2004; Barrett & Han, 2009; Brown, Boyd & Russell, 2011; C Boyd, pers. obs., 2011). The trigeminal foramen is entirely enclosed within the prootic, which is also seen in the basal iguanodontians Dryosaurus and Dysalotosaurus (Scheetz, 1999), and in some basal ceratopsians (e.g., Yinlong; C Boyd, pers. obs., 2011). The presence of a fossa on the medial surface of the prootic containing the foramina for CN VII and both branches of CN VIII is shared only with the basal iguanodontian Dysalotosaurus (Galton, 1989). In addition to these features, the lone autapomorphy of T. neglectus currently recognized is present on the braincase, as are four of the seven characters used to differentiate T. neglectus from T. assiniboiensis (see Emended diagnosis above), making this a critical region of the skull for evaluating the relationships of this species.

Relationship to Parksosaurus warreni

Some prior phylogenetic analyses of the relationships of Thescelosaurus positioned it in a monophyletic group with Parksosaurus (Weishampel et al., 2003; Boyd et al., 2009; Brown, Boyd & Russell, 2011). Alternatively, other analyses place Parksosaurus as the sister taxon to Gasparinisaura (Buchholz, 2002; Butler, Upchurch & Norman, 2008; Makovicky et al., 2011). This latter position was based upon character evidence that requires revision in light of new discoveries, recent preparation work on the holotype of Parksosaurus, and the new cranial data for T. neglectus described above. The analysis by Butler, Upchurch & Norman (2008) recovered three characters that unambiguously supported the monophyly of a Parksosaurus + Gasparinisaura clade. The first character (jaw joint strongly depressed ventrally, with more than 40% of the height of the quadrate below the level of the maxilla: Butler, Upchurch & Norman, 2008), is inaccurately scored for Parksosaurus based on misinterpretation of the holotype. The quadrate of Parksosaurus often is reconstructed as extremely dorsoventrally elongate, with the quadratojugal contacting the quadrate within the dorsal two-thirds of the quadrate shaft and a ventrally displaced jaw joint (e.g., Parks, 1926; Galton, 1973), a condition similar to that seen in Gasparinisaura (Coria & Salgado, 1996). However, on the holotype and only specimen of Parksosaurus, the left quadrate is displaced posteroventrally, rotated laterally about its long axis, and most of its jugal wing and ventrolateral margin are damaged and lost (C Boyd, pers. obs., 2011). The preserved, and newly prepared, morphology of the left quadrate of Parksosaurus exactly matches that of T. neglectus (Figs. 8C and 8D), including the presence of a foramen in the posterolateral side of the quadrate along the contact with the quadratojugal (Fig. 8G: C Boyd, pers. obs., 2011). Additionally, the distal end of the better preserved right quadrate is complete and exposed in posteromedial view, confirming the above observations regarding the length and morphology of the quadrate. The posterior portion of the lower jaw also is damaged in the holotype and displaced posteroventrally, enhancing the false impression that the jaw joint was positioned farther ventrally than it actually was. Thus, the position of the jaw joint and the morphology of the quadrate actually were similar to that of T. neglectus (Fig. 1).

The second character supporting a sister taxon relationship between Parksosaurus and Gasparinisaura in Butler, Upchurch & Norman (2008) is the presence of chevrons with anteroposteriorly expanded distal ends. That feature is present in both taxa, but it is also present in middle to distal caudals of Macrogryphosaurus, while the more anteriorly positioned chevrons of Macrogryphosaurus are relatively unexpanded (Calvo, Porfiri & Novas, 2007). This same variation was noted in Parksosaurus as well (Parks, 1926; C Boyd, pers. obs., 2011). In many specimens referred to Thescelosaurus, including the holotype and NCSM 15728, the middle to posteriorly positioned chevrons are damaged at their distal ends, making it impossible to determine if the same morphological variation is present. However, a second specimen of Thescelosaurus (referred based on the presence of the apomorphic morphology of the ribs outlined in the Systematic Paleontology section) held in the Timber Lake and Area Museum (TLAM.BA.2014.028.0001) preserves seven chevrons from the anterior and middle portions of the tail. The posterior-most preserved chevron in TLAM.BA.2014.028.0001 has a moderately anteriorly-posteriorly expanded distal end that matches the figured eighth chevron from the holotype of Parksosaurus (ROM 804: Parks, 1926: Fig. 4). This indicates that Thescelosaurus was at least polymorphic for this character, and may indicate a wider distribution of this character amongst neornithischians because few specimens include well-preserved chevrons from the middle or posterior portions of the tail.

The final character is the absence of a well-developed acromion process on the scapula. Although the acromion process is relatively pronounced in smaller specimens referred to Thescelosaurus (e.g., AMNH 5031: Galton, 1974b), in larger specimens of Thescelosaurus the acromion is less pronounced or nearly absent (e.g., MOR 989; NCSM 15728: C Boyd, pers. obs., 2011), suggesting that this character should be scored as polymorphic for Thescelosaurus. Given these observations, the characters outlined above do not provide strong support for the monophyly of a Parksosaurus + Gasparinisaura clade.

The monophyly of a Parksosaurus + Gasparinisaura clade in the study by Buchholz (2002) was supported by four characters, one of which is the shape of the chevrons (discussed above). The other three are a reduced or absent posterior process of the jugal, a long and thin anterior process of quadratojugal, and the presence of a large descending process of the quadratojugal (Buchholz, 2002). The first character is inaccurate for Parksosaurus because the posterior process of the jugal in Parksosaurus is elongate, forming nearly the entire ventral margin of the infratemporal fenestra, as in Thescelosaurus, and is incomplete ventrally so that its dorsoventral height cannot be determined. The scoring of the second character for Parksosaurus is suspect for two reasons. First, the quadratojugal is damaged and its exact dimensions cannot be determined. Second, in Thescelosaurus a long anterior process of the quadratojugal inserts medial to the jugal. Given the damage to the jugal in Parksosaurus, it cannot be determined if the increased anteroposterior length of the quadratojugal is merely the result of the quadratojugal being displaced, exposing the long anterior process. Finally, the third character is impossible to score with certainty in Parksosaurus owing to the incomplete preservation of the quadratojugal. Thus, reevaluation of the characters proposed to support a Parksosaurus + Gasparinisaura clade by Buchholz (2002) finds little support for this relationship.

Alternatively, Parksosaurus shares several characters in common with Thescelosaurus that are lacking in Gasparinisaura. Both Thescelosaurus and Parksosaurus possess in the posterolateral surface of the quadrate a foramen that passed medial to the quadratojugal (also seen in Haya and some iguanodontians: Norman, 2004; Makovicky et al., 2011). Thescelosaurus and Parksosaurus also possess ossified sternal ribs and intercostal plates (sensu Butler & Galton, 2008), both of which are also present in Hypsilophodon, Macrogryphosaurus, and Othnielosaurus (Butler & Galton, 2008; Boyd, Cleland & Novas, 2011) and the latter is present in Talenkauen (Boyd, Cleland & Novas, 2011). Finally, among neornithischians the fourth trochanter extends onto the distal half of the femur in Parksosaurus, Thescelosaurus, and Talenkauen (Gilmore, 1915; Galton, 1973; Novas, Cambiaso & Ambrosio, 2004). Additional characters supporting a close relationship between Thescelosaurus and Parksosaurus cannot be evaluated in Gasparinisaura, including the presence of a broad fossa at the base of the pterygoid wing of the quadrate (Fig. 8C). These observations suggest that Parksosaurus shared a closer relationship with Thescelosaurus than with Gasparinisaura, though the exact relationships of these taxa need to be reevaluated via a phylogenetic analysis incorporating these new character data.

Future directions in the study of Thescelosaurus

All three species of Thescelosaurus that are currently considered valid (T. assiniboiensis, T. garbanii, and T. neglectus) are from contemporaneous deposits, with the latter two present in the same formation (i.e., Hell Creek Formation; Boyd et al., 2009). The presence of multiple contemporaneous neornithischian taxa is not unique to the Western Interior Basin of North America during the late Maastrichtian. The taxa Agilisaurus louderbacki, Hexinlusaurus multidens, and Xiaosaurus dashanpensis are all from the Lower Shaximiao Formation of Sichuan Province, China (Barrett, Butler & Knoll, 2005). Similarly, two species of the basal iguanodontian taxon Zalmoxes are present during the early Maastrichtian in Romania (Weishampel et al., 2003; Panaiotu & Panaiotu, 2010). However, it is still imperative that the validity of all three species of Thescelosaurus be thoroughly evaluated. Although there is strong character evidence supporting the separation of T. neglectus and T. assiniboiensis (Brown, Boyd & Russell, 2011: this study), the same cannot be said for the fragmentary holotype of T. garbanii. The tentative retention of T. garbanii as a distinct taxon by Boyd et al. (2009) was based on review of the published data concerning the anatomy of that taxon because personal observation of the holotype material was not possible owing to the fact that the specimen was offsite at the time and unavailable for study. Clearly, it is crucial that the holotype material of T. garbanii be thoroughly reexamined and its validity confirmed once the material is available again for study.

Despite the excellent anatomical descriptions for T. neglectus (Gilmore, 1915; this study) and T. assiniboiensis (Brown, Boyd & Russell, 2011) now available, referral of additional specimens to individual species within Thescelosaurus remains problematic. This is largely a result of two factors. First, the fragmentary nature of the holotype of Thescelosaurus garbanii; and second, the lack of recognized postcranial autapomorphies for T. assiniboiensis and T. neglectus. With regards to the former factor, only the discovery of additional specimens clearly referable to T. garbanii can resolve the issue, assuming the validity of that species is upheld. The latter factor requires detailed examination not just of the holotypes of all three taxa, but of all well-preserved specimens referred to Thescelosaurus to elucidate any patterns of morphological variation within the taxon and disparity between species. Given the wide range of body sizes represented by specimens referred to Thescelosaurus (see Boyd et al., 2009: Fig. 2), the possible effects of ontogeny on postcranial (and cranial) skeletal morphology will need to be evaluated and taken into consideration. The results of such a study will be crucial to deciphering the life history strategy of Thescelosaurus, evaluating the ontogenetic status and comparability of specimens referred to Thescelosaurus, and identifying taxonomically informative differences between the postcranial skeletons of the three currently recognized species of Thescelosaurus.

Thanks first and foremost to the staff at the North Carolina Museum of Sciences for extensive access to NCSM 15728 over the years. This manuscript would not have been possible without their untiring assistance. Access to collections and specimens was generously provided by C Mehling and M Norell (AMNH), X Xing (IVPP), J Horner and B Baziak (MOR), V Schneider and D Russell (NCSM), KQ Gao (Peking University), CM Brown and D Evans (ROM), S Shelton and M Greenwald (SDSM), J Nelson, K Nelson, and M Marshall (TLAM), M Carrano and M Brett-Surman (USNM), S Sampson, M Getty, and R Irmis (UMNH), and D Brinkman and C Norris (YPM). P Brinkman (NCSM) collaborated in the preparation of NCSM 15728 and M Brown (VPL) provided instruction in the preparation of several new specimens of Scutellosaurus. The late B Alley discovered, collected, and prepared specimens TLAM.BA.2014.027.0001 and TLAM.BA.2014.028.0001, and then donated those specimens to the Timber Lake and Area Museum when he realized the benefit those specimens could provide to the scientific community. D Evans (ROM) provided information on the anatomy of ROM 804. DA Winkler provided anatomical information about the ‘Proctor Lake ornithopod.’ D Varricchio provided additional information on the anatomy of MOR 1636a. CM Brown provided photographs of several specimens referred to Thescelosaurus, information regarding orodromine material from the Dinosaur Park Formation, and details regarding the anatomy of Parksosaurus. PM Galton provided encouragement and enlightening conversations over the years regarding basal ornithischian taxa. TP Cleland provided select photographs of NCSM 15728 and USNM 7758. S Nesbitt provided access to photographs of several basal ornithischian taxa from South Africa. EL Schmidt provided illustrations of NCSM 15728. C Gardner generously donated her time and talent to design the outlines used in Figs. 1–4. I am indebted to S Masters for bringing the neornithischian material from the Kaiparowits Formation of Utah to my attention and A Titus of the Utah Bureau of Land Management for assistance in locating additional specimens referable to that new taxon. The Polyglot Paleontologist website provided access to multiple translations of important scientific research papers on ornithischian dinosaurs. CJ Bell, D Cannatella, JA Clarke, TP Cleland, A DeBee, DR Eddy, ML Householder, P Makovicky, S Nesbitt, and T Rowe provided thoughtful comments on previous drafts of this manuscript that greatly improved its quality.

Institutional Abbreviations

AMNH American Museum of Natural History, New York, New York, USA

CMN Canadian Museum of Nature, Ottawa, Ontario, Canada

IVPP Institute of Vertebrate Paleontology and Paleoanthropology, Chinese Academy of Sciences, Beijing, China

MOR Museum of the Rockies, Bozeman, Montana, USA

NCSM North Carolina Museum of Natural Sciences, Raleigh, North Carolina, USA

PKUP Peking University, Beijing, China

ROM Royal Ontario Museum, Toronto, Ontario, Canada

RSM Royal Saskatchewan Museum, Regina, Saskatchewan, Canada

SDSM South Dakota School of Mines and Technology, Rapid City, South Dakota, USA

TLAM.BA Timber Lake and Area Museum (Bill Alley Collection), Timber Lake, South Dakota, USA

UMNH VP Utah Museum of Natural History, Salt Lake City, Utah, USA

USNM Smithsonian Institution, National Museum of Natural History, Washington, D. C., USA

VPL Vertebrate Paleontology Lab, The University of Texas at Austin, Austin, Texas, USA

YPM Yale Peabody Museum, New Haven, Connecticut, USA.

Additional Information and Declarations

Competing Interests

Author Contributions

The author declares there are no competing interests.

Clint A. Boyd conceived and designed the experiments, performed the experiments, analyzed the data, contributed reagents/materials/analysis tools, wrote the paper, prepared figures and/or tables, reviewed drafts of the paper.

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
