# Peer review of "The cranial anatomy of the neornithischian dinosaur Thescelosaurus neglectus"

_PeerJ, doi:10.7717/peerj.669_

## Round 0.1 · original submission · Minor Revisions

· Academic Editor

Minor Revisions

This is truly a well-written, beautifully illustrated, and comprehensive manuscript. The comments from the reviewers are generally fairly minor, and my hope is that they should be relatively easy to incorporate. Some additional notes and comments follow.

- The only major thing I find lacking in this description: measurements. At a minimum, some basic lengths and widths (e.g., dentary length; frontal length and width; total skull length; etc.) should be provided for comparison. A comprehensive set of measurements for the cranial elements would be even better. Size and proportions are an important part of the anatomy!
- Can you say anything about the quality of the data used to reconstruct the bones from CT scans? What was the voxel size (not just slice spacing and thickness)? Are the data reposited anywhere (whether at an online database or physically at the museum)? Even for the best dataset, CT scan reconstructions are interpretations by the software and the technician, so it is important to be able to verify them in some way. This was also noted by one of the reviewers.
- Does the TLAM specimen have a number? Is it formally accessioned there? As I recall, the specimen was in a local rancher's personal collection for many years, and then exhibited at the TLAM, prior to the rancher's passing. Was the specimen ever formally donated? I think this may be the issue to which one of the reviewers refers--based on the my experience, the TLAM is a legitimate public institution, but it would be a good idea to confirm that the specimen is indeed accessioned there.
- The figures are beautifully detailed and comprehensive. This alone will be an incredible resource! The only minor, optional, addition I can recommend would be to include an outline drawing to accompany the views in Figures 2-4. Visually, this would allow you to remove the labels from over the photograph of the skull, and it would also make it easier for the reader to distinguish between sutures, cracks, etc. This would help address some of the requests for figure clarification by the reviewers.
- p. 85: Additionally, better stratigraphic placement of Thescelosaurus specimens might be informative, to see if any morphological variation is plausibly the result of evolutionary change over time.

Figure 16 - typo for “left”?

·

Basic reporting

No Comments

Experimental design

No Comments

Validity of the findings

No Comments

Additional comments

This is an excellent paper, with great figures of great new material of an important taxon. The anatomical description of the skull of Thescelosaurus will be very useful for other researchers, such as myself, working on the systematics and phylogeny of Neornithischia and its subclades. There are some points on which I think the paper could be improved.

Systematic Paleontology

You should note that the strict reduced consensus tree in the latest iteration of Butler et al.’s (2008) analysis placed Thescelosaurus in Ornithopoda, even when a broad sample of non-ornithopod ornithischians was included (Han et al. 2012).

Since the referral of NCSM 15728 to T. neglectus is so important for systematic studies, the features linking it, the type series, and the Timber Lake specimen should be explained in more detail.

Description of the Skull of NCSM 15728

You make extensive comparisons with other ornithischian taxa. However, I think you should include comparisons with other skulls of Thescelosaurus sp., especially SDSM 7210 and MOR 979, insofar as they are available.

Future directions in the study of Thescelosaurus

Yandusaurus hongheensis is actually from the Upper Shaximiao Formation, and was not a contemporary of those three other taxa (Barrett et al. 2005).

You should omit the example of Orodromeus makelai and the new Kaiparowits taxon. They are not from the same unit, in contrast to the three Chinese taxa from the Lower Shaximiao Formation, the two species of Zalmoxes from the Sânpetru Formation, and T. neglectus and T. garbanii from the Hell Creek.

Zalmoxes robustus and Z. shqiperorum are early Maastrichtian in age (Panaiotu and Panaiotu 2010).

If the Timber Lake specimen is functioning as a ‘Rosetta Stone’ by which you are able to refer NCSM 15728 to T. neglectus, why can it not do the same for those other specimens with good skulls? MOR 979 also has the tarsus, and so can be compared with both the Timber Lake specimen and T. garbanii.

·

Basic reporting

This is an excellent descriptive paper - one of the best I've seen in a long time and a genuine pleasure to read. Detailed, thorough, well argued and very clearly written, with a good level of comparative data. As a result, it will be an exceptionally useful contribution to workers in the field. The author provides all relevant background and his coverage of the literature is adequate, with only a smattering of very recent references missing. There are a few minor typos and formatting errors in the text that require correction, but otherwise the text presentation is generally excellent. The CT images are very useful and clear, as are the majority of the photographic figures: however, several of the images - especially those of the entire skull and the tooth rows - are somewhat washed out and slightly fuzzy and the author should be encouraged to provide sharper images with better contrast, if possible.

Experimental design

To my knowledge this paper presents work that is original and that has not been published elsewhere. The aim is simple (description of an interesting new specimen) and the author achieves that aim well. I have one concern in this area: some of the specimens cited for comparative purposes appear to be in private collections. References to these privately-owned specimens should be removed as there is no guarantee that observations made on these can be repeated by others in the future.

Validity of the findings

The descriptions, interpretations and conclusions of the author generally seem fine to me - I have very few points of disagreement and these are limited to a small number of very specific comments on particular anatomical structures. These are all minor and I have recorded these (along with some typographic errors) on the attached annotated version of the MS. This is a thoroughly competent and excellent piece of work.

Additional comments

No Comments

Reviewer 3 ·

Basic reporting

The paper adheres to all PeerJ policies and templates, is well-written, and includes sufficient background and introduction (including making extensive comparisons with a a very wide range of other ornithischian taxa) . Figures are generally excellent (although some images could be made clearer - see detailed comments in amended PDF). The article is a very thorough and self-contained description of the cranium of Thescelosaurus.

Experimental design

The article represents primary research and is very thorough; this is an anatomical description of very high quality.

Validity of the findings

The specimen on which the article is based is housed in a public institution and accessioned, and is therefore available to other researchers to verify the analysis. My only concern here is: where will the CT scans be deposited? Will they be made available to other researchers?

Additional comments

This is an excellent, very complete and thorough description of the skull of an important and long-neglected dinosaur taxon. The use of both actual specimens and CT scans in the description is admirable and the figures are generally excellent. The description of the braincase is especially well-written, complete and interesting. There are only some minor errors (throughout the article) that require correction - see sticky notes in the attached PDF. Overall, an excellent and very valuable contribution to the literature.

One note - I would ask the author to specify, at some point in the article, when and how the CT scan data will be made available to other researchers?

Annotated reviews are not available for download in order to protect the identity of reviewers who chose to remain anonymous.

---

## Round 0.2 · Minor Revisions

· Academic Editor

Minor Revisions

Thank you for your detailed incorporation of and response to the comments from the reviewers and me. I think the manuscript is nearly ready to go; the only remaining issue concerns inclusion of some measurements.

The explanation for why some measurements (e.g., total skull length) are not suitable for the specimens under discussion makes sense to me. I also do not think it is desirable to force you to fly out to North Carolina just for a measurement table. Thus, it is sufficient to include just those measurements that you already have in hand (e.g., width of frontals, height of quadrate, length of maxillary tooth row, total length of premaxilla, total length of dentary, oral margin of predentary; and right quadrate length from the TLAM specimen). These are useful metrics, and should provide much of what interested researchers will need.

While you are revising the manuscript, there is a minor typo on p. 13 that can be fixed -- "putitive" should be spelled as "putative".

I look forward to seeing your revisions, and should be able to turn them around for a final decision very quickly.

---

## Round 0.3 · accepted · Accept

· Academic Editor

Accept

Thank you for the quick turn-around on the final round of comments. This manuscript will be an excellent (and I am sure frequently used) addition to the literature.